# A chemical biology toolbox to study protein methyltransferases and epigenetic signaling

Sebastian Scheer [1], Suzanne Ackloo [2], Tiago S. Medina[3], Matthieu Schapira [2,4], Fengling Li[2], Jennifer A. Ward [5,6], Andrew M. Lewis [5,6], Jeffrey P. Northrop[1], Paul L. Richardson [7], H. Ümit Kaniskan [8], Yudao Shen[8], Jing Liu [8], David Smil [2], David McLeod [9], Carlos A. Zepeda-Velazquez[9], Minkui Luo [10,11], Jian Jin [8], Dalia Barsyte-Lovejoy [2], Kilian V.M. Huber [5,6], Daniel D. De Carvalho[3,12], Masoud Vedadi[2,4], Colby Zaph [1], Peter J. Brown [2] & Cheryl H. Arrowsmith[2,3,12]

Protein methyltransferases (PMTs) comprise a major class of epigenetic regulatory enzymes with therapeutic relevance. Here we present a collection of chemical probes and associated reagents and data to elucidate the function of human and murine PMTs in cellular studies. Our collection provides inhibitors and antagonists that together modulate most of the key regulatory methylation marks on histones H3 and H4, providing an important resource for modulating cellular epigenomes. We describe a comprehensive and comparative characterization of the probe collection with respect to their potency, selectivity, and mode of inhibition. We demonstrate the utility of this collection in CD4[+] T cell differentiation assays revealing the potential of individual probes to alter multiple T cell subpopulations which may have implications for T cell-mediated processes such as inflammation and immuno-oncology. In particular, we demonstrate a role for DOT1L in limiting Th1 cell differentiation and maintaining lineage integrity. This chemical probe collection and associated data form a resource for the study of methylation-mediated signaling in epigenetics, inflammation and beyond.

[1] Infection and Immunity Program, Monash Biomedicine Discovery Institute, Department of Biochemistry and Molecular Biology, Monash University, Clayton, VIC 3800, Australia. [2] Structural Genomics Consortium, University of Toronto, Toronto, ON M5G 1L7, Canada. [3] Princess Margaret Cancer Centre, University Health Network, Toronto, ON M5G 2M9, Canada. [4] Department of Pharmacology and Toxicology, University of Toronto, Toronto, ON M5S 1A8, Canada. [5] Structural Genomics Consortium, University of Oxford, Oxford OX3 7DQ, UK. [6] Target Discovery Institute, Nuffield Department of Medicine, University of Oxford, Oxford OX3 7FZ, UK. [7] AbbVie Inc., 1 North Waukegan Rd, North Chicago, IL 60064, USA. [8] Mount Sinai Center for Therapeutics Discovery, Departments of Pharmacological Sciences and Oncological Sciences, Tisch Cancer Institute, Icahn School of Medicine at Mount Sinai, New York, NY 10029, USA. [9] Ontario Institute for Cancer Research, Toronto, ON M5G 0A3, Canada. [10] Chemical Biology Program, Memorial Sloan Kettering Cancer Center, New York, NY 10065, USA. [11] Program of Pharmacology, Weill Cornell Medical College of Cornell University, New York, NY 10021, USA. [12] Department of Medical Biophysics, University of Toronto, Toronto, ON M5G 1L7, Canada. Correspondence and requests for materials should be addressed to C.Z. (email: colby.zaph@monash.edu) or to P.J.B. (email: peterj.brown@utoronto.ca) or to C.H.A. (email: cheryl.arrowsmith@uhnresearch.ca)

Epigenetic regulation of gene expression is a dynamic and reversible process that establishes and maintains normal cellular phenotypes, but contributes to disease when dysregulated. The epigenetic state of a cell evolves in an ordered manner during cellular differentiation and epigenetic changes mediate cellular plasticity that enables reprogramming. At the molecular level, epigenetic regulation involves hierarchical covalent modification of DNA and the histone proteins that package DNA. The primary heritable modifications of histones include lysine acetylation, lysine mono-, di-, or tri-methylation, and arginine methylation. Collectively these modifications establish chromatin states that determine the degree to which specific genomic loci are transcriptionally active[1].

Proteins that read, write, and erase histone (and non-histone) covalent modifications have emerged as druggable classes of enzymes and protein–protein interaction domains[2]. Histone deacetylase (HDAC) inhibitors and DNA hypomethylating agents have been approved for clinical use in cancer and more recently clinical trials have been initiated for antagonists of the BET bromodomain proteins (which bind to acetyllysine on histones), the protein methyltransferases EZH2, DOT1L, and PRMT5, and the lysine demethylase LSD1[3]. The development of this new class of epigenetic drugs has been facilitated by the use of chemical probes to link inhibition of specific epigenetic protein targets with phenotypic changes in a wide variety of disease models, thereby supporting therapeutic hypotheses[4].

Methylation of lysine and arginine residues in histone proteins is a central epigenetic mechanism to regulate chromatin states and control gene expression programs[5–7]. Mono-, di-, or tri-methylation of lysine side chains in histones can be associated with either transcriptional activation or repression depending on the specific lysine residue modified and the degree of methylation. Arginine side chain methylation states include mono-methylation and symmetric or asymmetric dimethylation (Fig. 1a). In humans two main protein families carry out these post-translational modifications of histones. The structurally related PR and SET domain containing enzymes (protein lysine methyltransferases (PKMT)) methylate lysine residues on histone "tails", and the dimeric Rossman fold protein arginine methyltransferase (PRMT) enzymes modify arginine. DOT1L has the Rossman fold, but is a monomer and modifies a lysine on the surface of the core histone octamer within a nucleosome (as opposed to the disordered histone tail residues). Many of these proteins also methylate non-histone proteins, and even less is known about non-histone methylation signaling[8,9].

Here we describe a resource of chemical probes and related chemical biology reagents to study the cellular function of PMTs, and link inhibition of select PMTs to biological mechanisms and therapeutic potential. We summarize the key features of each probe including its potency, selectivity, biochemical and cellular activity, and mode of action for a comprehensive data resource for the collection. We also describe a control compound to use for each probe that is structurally similar but inactive on the enzyme. Furthermore, a set of affinity reagents derived from each chemical probe is presented and their use is exemplified in cellular selectivity and chemical proteomics experiments. Finally, we use the entire collection of chemical probes to examine the effects of inhibition of individual PMTs on the ability of naïve T cells to differentiate into effector T cell lineages. These data reveal links between epigenetic regulators and T cell biology in both humans and mice, and in so doing demonstrate how the chemical probe collection may be used to explore the biology of these PMTs.

## Results

### A collection of inhibitors of the major histone methyl marks.
Table 1 lists chemical probes for human protein methyltransferases (PMTs) and key characteristics of their activity. This collection provides significant coverage of the human histone lysine and arginine methyltransferase phylogenetic trees (Fig. 1a), but more importantly includes modulators of the major regulatory histone methylation marks (Fig. 1b). Key among these are H3K9me2, H3K27me3, H3K79me2, and H4K20me2/3, each of which are written exclusively by G9a/GLP, PRC2 complex (via EZH1/2), DOT1L and SUV420H1/2 enzymes, respectively. As such, the respective chemical probes for these enzymes are able to reduce the global levels of their resultant mark in cells as measured by western blot, in-cell western, or immunofluorescence assays[10–13]. Other histone marks such as H3K4me1/2/3 are written by multiple enzymes. Thus, a chemical probe such as OICR-9429, which disrupts the MLL1 complex, is more likely to have specific effects only at loci targeted by MLL1, and not necessarily all methylated H3K4 loci[14]. PMTs also have many non-histone targets, including transcription factors such as p53[15], estrogen receptor[16,17], and cytosolic signaling factors such as MAP3K2[18]. In addition to histone modification, arginine methylation plays an important role in the function of RNA-binding proteins, ribosome biogenesis and splicing[19–21]. Thus, this collection of chemical probes constitutes a broad resource to link enzyme activity to a wide range of epigenetic and non-epigenetic methylation-mediated signaling pathways and biology.

### PMT chemical probes are potent, selective and cell-penetrant.
The development of these chemical probes was guided by principles practiced in the pharmaceutical industry to test the link between a specific protein and a putative biological or phenotypic cellular trait[4,22–24]. A useful chemical probe should be reasonably potent in cells, selective for the intended target protein, and free of confounding off-target activities.

The chemical probes described here were each discovered using a biochemical enzymatic assay for the respective recombinant protein, or in some cases the relevant recombinant multiprotein enzyme complex, where the probe has been demonstrated to have an on-target potency with $IC_{50} < 100$ nM (Table 1). Each probe was evaluated in a customized cellular assay that tested the ability of the probe to reduce the level of methylation of its substrate in cells. All probes have significant, on-target cellular activity at 1 μM making them useful tools for cellular studies. Importantly these chemical probes are highly selective for their target protein (Fig. 1c); each has been screened against a collection of up to 34 human SAM-dependent histone, DNA and RNA methyltransferases. The chemical probes within the lysine methyltransferase family are highly selective with measurable cross reactivity only seen between very closely related proteins such as G9a and GLP, or SUV420H1 and SUV420H2. Selectivity within the PRMT family, however, is more difficult to achieve, and a greater degree of cross reactivity is seen in this subfamily. The probes had minimal or no off-target activities when screened against a panel of 119 membrane receptors and ion channels, and kinases (Supplementary Table 1).

Importantly, most chemical probes are accompanied by a structurally similar control compound that has similar physicochemical properties but is inactive or much less active on its target enzyme (Table 1, Supplementary Tables 1 and 2). These inactive compounds are to be used alongside the active probe to control for unanticipated off-target activity of their common chemical scaffold. In cases where an appropriate control molecule could not be identified, an alternative strategy is to use multiple chemical probes with different chemotypes that inhibit the same target (Table 1 and Supplementary Table 2). Many of the PMTs discussed here have two or more chemical probes with different chemotypes, mechanisms of action, and/

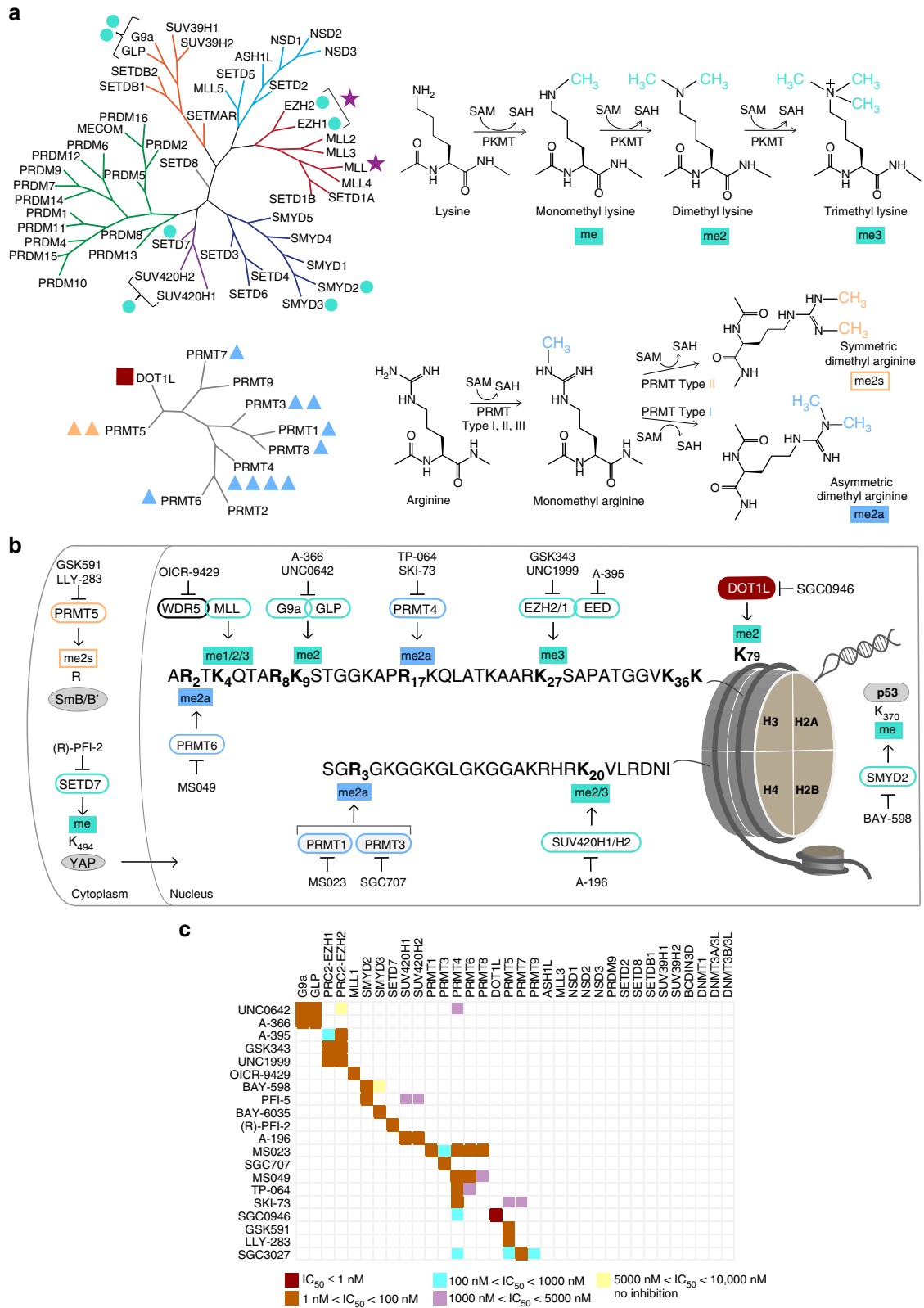

or potency profile. By using multiple diverse chemical probes that inhibit the same target, the user can build confidence in the link between cellular phenotype and inhibition of a specific target.

**Multiple mechanisms to inhibit PMTs.** The methyl-donating cofactor *S*-adenosylmethionine (SAM) and methyl-accepting substrate of protein methyltransferases bind at juxtaposed but distinct sites that can each be targeted by small molecule

**Fig. 1** Summary of chemical probes. **a** Phylogenetic trees of human PR and SET domain lysine methyltransferases (upper tree), and the β-barrel fold enzymes (lower tree). Trees are annotated to show chemical probes in this collection that inhibit PKMTs (turquoise circle), a Rossman fold PKMT (dark red square), monomethyl and asymmetric dimethyl PRMTs (blue triangle), symmetric dimethyl PRMTs (orange triangle); and methyltransferase protein complexes (purple star). The number of annotations adjacent to each target is equal to the number of chemical probes for that target. **b** Detailed coverage of the major histone H3 and H4 methyl marks modulated by this collection of chemical probes. The methylated lysine (K) and arginine (R) residues are annotated in bold font. The PMTs that write the marks are shown with green (PKMTs) or blue (PRMTs) borders, along with the chemical probes that inhibit these PMTs. Also included are non-histone substrates (gray ovals) of PRMT5, SETD7, and SMYD2. **c** Selectivity of each chemical probe has been assessed against 34 SAM-dependent methyltransferases. SKI-73 and SGC3027 are pro-drugs and selectivity was determined on the respective active components. See also Supplementary Tables 1-3. SAM S-adenosylmethionine, SAH S-adenosylhomocysteine, me methyl, me2a asymmetric dimethyl, me2s symmetric dimethyl, me2/3 di- and tri-methyl marks are written by the same enzyme

**Table 1 Summary of chemical probes and their chemotype-matched controls for protein methyltransferases. Related to Fig. 1a–c, Supplementary Figs. 1 and 2**

| Target | Probe | In vitro IC$_{50}$ or K$_d$ (nM) | Cell line: assay | IC$_{50}$ (nM) | Control |
|---|---|---|---|---|---|
| WDR5[a] | OICR-9429[14] | 64 | HEK293: disrupt interaction of WDR5[b] with MLL1 & RbBP5 | 223 & 458 | OICR-0547 |
| EED[a] | A-395[35] | 34 | RD: ↓ H3K27me3 | 90 | A-395N |
| EZH2 | GSK343[63] | 4 | HCC1806: ↓ H3K27me3 | 250 | |
| EZH2,H1 | UNC1999[11] | 10,45 | MCF10A: ↓ H3K27me3 | 124 | UNC2400 |
| DOT1L | SGC0946[12] | 0.3 | MCF10A,A431: ↓ H3K79me2 | 10,3 | SGC0649 |
| G9a & GLP | A-366[26] | 4 & 38 | PC3: ↓ H3K9me2 | 300 | Use UNC0642 |
| G9a & GLP | UNC0642[10] | <3 | PC3: ↓ H3K9me2 | 130 | Use A-366 |
| SUV420H1/2 | A-196[13] | 21 | U2OS: ↓ H4K20me2/3 | 262/370 (SUV420H1) | A-197, SGC2043 |
| SMYD2 | BAY-598[29] | 27 | HEK293, SMYD2[b]:↓ p53K370me | 58 | BAY-369 |
| SMYD3 | BAY-6035[c] | 88 | HeLa, SMYD3[b]: ↓ MEKK2K260me3 | 70 | BAY-444 |
| SETD7 | (R)-PFI-2[27] | 2 | MEFs and MCF7: ↑ nuclear YAP | sub-μM | (S)-PFI-2 |
| Type 1 PRMTs | MS023[37] | 30,8 (PRMT1,6) | MCF7 (PRMT1), HEK293 (PRMT6[b]): ↓ H4R3me2a, ↓ H3R2me2a | 9,56 (PRMT1,6) | MS094 |
| PRMT3 | SGC707[34] | 31 | HEK293, PRMT3[b]: ↓ H4R3me2a | 91 | XY1 |
| PRMT4 | TP-064[64] | <10 | HEK293: ↓ Med12me2a, ↓ Baf155me2a | 43, 340 | TP-064N |
| PRMT4 | SKI-73[d,e] | 11 | MCF-7: ↓ Med12me2a | 540 | SKI-73N |
| PRMT4,6 | MS049[65] | 44,63 | HEK293: ↓ Med12me2a (PRMT4), ↓ H3R2me2a (PRMT6[b]) | 1400, 970 | MS049N |
| PRMT5 | GSK591[28] | 11 | Z138: ↓ SmD3me2s | 56 | SGC2096 |
| PRMT5 | LLY-283[31] | 22 | MCF7: ↓ SmBB'Rme2s | 30 | LLY-284 |
| PRMT7 | SGC3027[d,f] | <2.5 | C2C12, PRMT7[b]: ↓ HSP70R469me | 2400 | SGC3027N |

[a]Subunits required for activity of their respective enzyme complexes
[b]Cellular assay utilizes exogenous, transfected target
[c]https://www.thesgc.org/chemical-probes/BAY-6035
[d]SKI-73 and SGC3027 are pro-drugs that are converted to the active compound by reductases in the cell. In vitro data are shown for the active component
[e]http://www.thesgc.org/chemical-probes/SKI-73
[f]https://www.thesgc.org/chemical-probes/SGC3027
↓ = decrease; ↑ = increase

inhibitors (Fig. 2). Early efforts to inhibit PMTs focused on targeting the SAM binding site in analogy to targeting the ATP binding site of kinases[25]. However, it has been challenging to identify cell-penetrant compounds that bind to the polar SAM binding pocket and no universal methyltransferase inhibitor scaffold or warhead has yet been identified.

The most frequent mode of inhibition for PMTs is binding of the probe within the substrate pocket, thereby preventing substrate binding (Fig. 2a)[13,26–29]. The high selectivity profile of the SET-domain chemical probes is likely related to the high degree of substrate selectivity of these enzymes[30]. These substrate-competitive probes also take advantage of the structural malleability of the substrate-binding groove to remodel the substrate-binding loops for optimal fit (Fig. 2a, gray contours). Interestingly, binding of many of these substrate competitive probes is also dependent on cofactor SAM, which in some cases directly interacts with the probe molecule, and also is known to help stabilize formation of the substrate-binding pocket of SET domain proteins[30]. These contributions of SAM can be a confounding factor when interpreting enzyme inhibitory kinetic data for these probes. In Fig. 2 we have focused on the mechanism of action of each probe based on the structures of their complexes with their target enzyme. The potent SAM competitive inhibitors, SGC0946[12] (Fig. 2b) and LLY-283[31], have adenosine-like moieties with hydrophobic substituents replacing the methionine of SAM, while UNC1999[11] and GSK343[32] have a

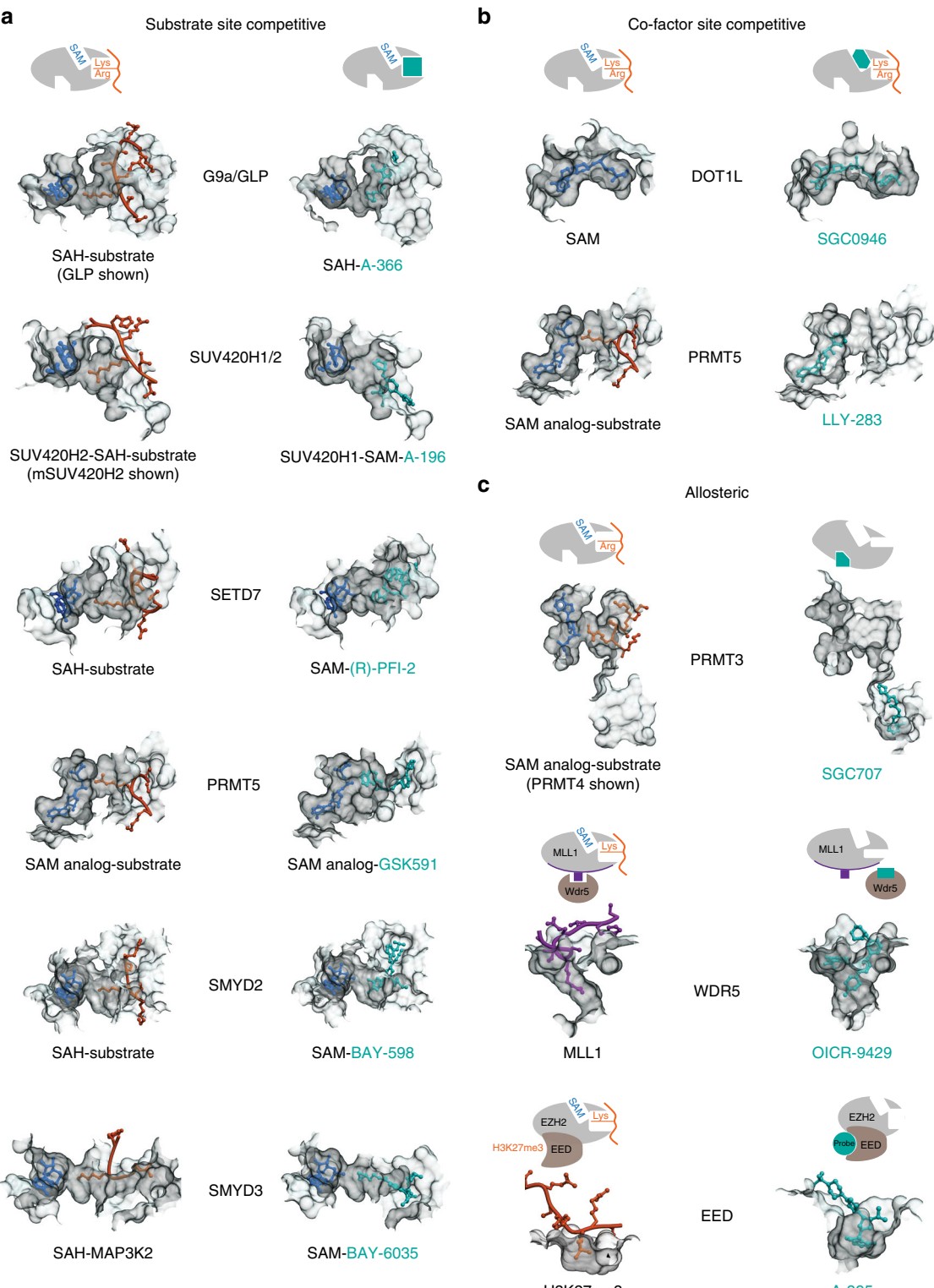

**Fig. 2** Structural mechanisms of PMT inhibition by chemical probes. **a** Inhibitors of G9a (PDB: 3HNA (GLP with H3K9 substrate) and 4NVQ (A-366));
SUV420H2 (PDB: 4AU7 (mSUV420H2 with H4 substrate) and SUV420H1 (PDB: 5CPR (A-196)); SETD7 (PDB: 1O9S (H3 substrate) and 4JLG (PFI-2));
PRMT5 (PDB: 4GQB (H4 substrate) and 5C9Z (GSK591)); SMYD2 (PDB: 3TG5 (p53 substrate) and 5ARG (BAY-598)); SMYD3 (PDB: 5EX0
(MAP3K2 substrate) and (BAY-6035)) all bind in the substrate (peptide) binding pocket. **b** SGC0946 binds in the SAM-binding pocket of DOT1L thereby
preventing cofactor binding (PDB ID: 3QOW (SAM), 4ER6 (SGC0946)). LLY-283 also occupies the SAM-binding pocket of PRMT5-MEP50 complex (PDB
ID: 4GQB (SAM analog) and 6CKC (LLY-283)). **c** Three distinct modes of allosteric inhibition of protein methyltransferases. SGC707 binds to PRMT3 in an
allosteric site that prevents productive formation of the enzyme's activation helix (PDB ID: 4RYL). OICR-9429 binds to WDR5 and inhibits MLL1 activity by
disrupting WDR5-MLL1-RBBP5 complex (PDB ID: 4QL1). A-395 binds to the EED subunit of the PRC2 complex thereby preventing binding of activating
peptides (PDB ID: 5K0M)

pyridone-based scaffold that binds in and near the unique pocket of the PRC2 multiprotein complex[33].

A third class of inhibitors has allosteric mechanisms that can induce long-range structural perturbations, or take advantage of sites of protein-protein interactions within multi-subunit PMT complexes. The allosteric inhibitor SGC707[34] occupies a pocket that is 15Å away from the site of methyl transfer, but prevents formation of the catalytically competent conformation of the PRMT3 dimer (Fig. 2c, top panel). OICR-9429[14] and A-395[35] are antagonists that make use of the peptide binding pocket in the essential WD40 subunits (WDR5 and EED) of the multiprotein MLL1 and PRC2 complexes, respectively (Fig. 2c, middle and bottom panels). OICR-9429 binds in the central pocket of WDR5 preventing the latter's interactions with MLL and histone peptides, resulting in diminished methyltransferase activity of the complex[14]. A-395 binds in the analogous pocket of EED, thereby preventing its interaction with trimethylated peptides that activate the PRC2 holoenzyme[35]. Both chemical probes burrow into the central cavity of their respective WD40 protein targets occupying a remodeled binding pocket that is significantly larger than that of the peptides which they replace. Thus, a common feature revealed by the co-crystal structures of these probes with their target protein is the significant remodeling of the substrate or allosteric binding sites to selectively accommodate each probe molecule.

A caveat of targeting scaffolding subunits of chromatin complexes is that such proteins are often components of multiple complexes, with diverse functions in cells. For instance, WDR5 interacts with at least 64 different proteins, including the oncoprotein c-MYC, and disrupting the WDR5 protein interaction network has diverse functional consequences beyond the MLL complex. In this regard, chemical probes are ideal tools to investigate the biochemical and cellular outcome of targeting a specific protein interaction interface. Taken together, this collection of chemical probes reveals the wide array of mechanisms by which PMTs may be inhibited.

**Reagents for specificity and protein interactome studies.** Chemical proteomics using affinity reagents derived from chemical probes represents a powerful approach to investigate the molecular target profile of chemical probes[36] and to elucidate protein interaction networks in multiple cell types. In order to facilitate such studies ten affinity reagents were synthesized for seven targets. Of these reagents, three have been previously used to verify the selectivity of their respective chemical probe for the intended target protein[11,14,27], and for an additional five probes (for four targets) we indicate here the recommended site of derivatization (Supplementary Table 2). These suggestions are based on empirical data obtained during each probe discovery program, including target/probe co-crystal structures and chemical structure-activity-relationships (SAR). A further three affinity reagents were used to generate cellular selectivity data for the DOT1L inhibitor SGC0946, the EED antagonist A-395, and the Type I PRMT inhibitor MS023[37] (Fig. 3). The EED and Type I PRMT affinity reagents specifically enriched their cognate targets from cell lysates as demonstrated by immunoblotting (Supplementary Fig. 1). This enrichment could be competed with the respective underivitized free chemical probe suggesting a specific interaction not affected by immobilization. Profiling of the pan Type I PRMT inhibitor MS023 in HEK293 cell lysates using label-free quantification (LFQ) protein mass spectrometry showed engagement of several PRMTs including PRMT1, 3, 4, and 6 and their respective binding partners (Fig. 3a, d, and Supplementary Data 1). PRMT8, which MS023 is known to inhibit, was not detected, consistent with lack of expression in HEK293 cells. The

negative control compound MS094 had no effect on PRMT target engagement or interactome. Investigation of the EED antagonist A-395 in G401 cell lysates revealed several well-known PRC2 complex members which were co-purified with the cognate target EED (Fig. 3b, e, and Supplementary Data 2) similar to results obtained with a structurally distinct EED antagonist[38]. As expected, pre-treatment with the negative control A-395N even at high concentration did not affect binding of EED and the other PRC2 complex members to the A-395 affinity matrix, confirming its applicability as an additional tool for EED and PRC2. Chemoproteomic analysis of the DOT1L inhibitor SGC0946 in Jurkat cell lysates indicated that the probe is highly specific for DOT1L, confirming earlier results obtained with HL-60 cells[39] (Fig. 3c and Supplementary Data 3).

**Approaches to use the collection.** Each chemical probe can be used to investigate the effects of inhibiting the respective PMT in a biochemical or cellular assay. Thus, each probe and its control (s) will be valuable tools in hypothesis-driven research projects. We envision, however, that these tools will be equally valuable in prospective, hypothesis-generating discovery screens to identify specific PMTs whose inhibition is linked with a certain phenotype. To date, much of the research using epigenetic inhibitors has focused on cancer research, with inhibitors of several PMT targets progressing into clinical trials. While these oncology studies are of great interest and medical importance, we sought to systematically investigate the landscape of PMT inhibition in a non-cancer setting, namely CD4$^+$ T helper (Th) cell differentiation.

**Regulation of Th cell differentiation by PMTs.** Cellular differentiation is guided by chromatin-mediated changes in gene expression in response to extracellular cues. In the immune system, naïve Th cells can adopt a wide range of cellular fates depending upon the external signals received during activation by antigen-presenting cells[40]. For example, interleukin (IL)-12 promotes the development of Th1 cells that express the transcription factor T-bet (Tbx21) and produce interferon (IFN)-γ, while activation in the presence of IL-4 leads to Th2 cells that express GATA-3 and secrete IL-4, IL-5 and IL-13. In addition, Th17 (RORγt-expressing and IL-17A-producing) and induced regulatory T (Treg) cells (FOXP3-expressing) develop from naïve Th cells in the presence of TGFβ and IL-6 or TGFβ alone, respectively. The magnitude of the immune response to a given stimulus is dependent on the abundance and relative proportions of Th subtypes, and dysregulation of the balance among Th subtypes contributes to diseases such as autoimmunity and cancer[41]. Previous studies using gene-deficient mice have identified central roles for PMTs including MLL1[42], G9a[43,44], and EZH2[45,46] in Th cell differentiation and stability.

We tested the effect of inhibiting each methyltransferase on Th cell differentiation, focusing first on IFN-γ-producing Th1 cells (Fig. 4a, b, and Supplementary Figs. 2 and 3). Isolated naïve CD4$^+$ Th cells from the spleen and peripheral lymph nodes of mice with a fluorescent reporter of IFN-γ expression[47] (IFN-γ-YFP mice) were activated for 4 days under Th1 polarizing conditions in the presence of each probe and its respective control compound (when available). Treatment with UNC1999 or GSK343 which target the catalytic subunits of PRC2 (EZH1/2), (but not the control compound UNC2400) significantly increased the expression and production of the signature cytokine IFN-γ under Th1 polarizing conditions. These results are consistent with previous studies showing that T cell-specific deletion of EZH2 resulted in enhanced Th1 cell differentiation in mice[45,46,48]. We also observed a significant increase of IFN-γ producing CD4$^+$

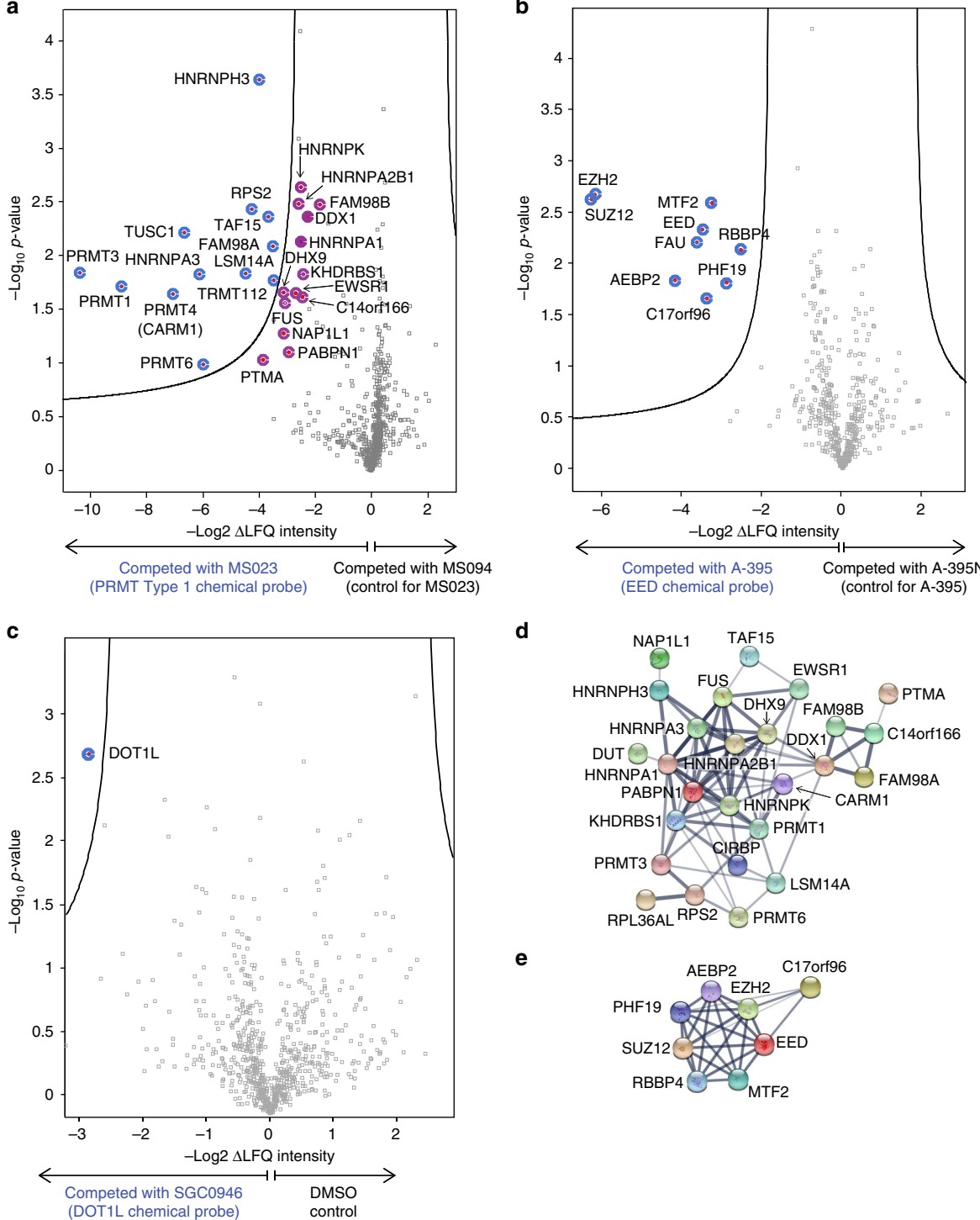

**Fig. 3** Affinity Reagents for Chemoproteomics. **a** Volcano plot of MTM7172-enriched proteome (labeled blue) from HEK293 cell lysate, with targets significantly (FDR=0.05, S0 = 0.2) competed by 20 μM MS023 with respect to 20 μM MS094 negative control. Proteins marked in purple indicate known interactors of the identified direct targets of MS023 close to the significance threshold. **b** Volcano plot of (A-395)-NH₂-enriched proteome (labeled blue) from G401 cell lysate, with targets significantly (FDR = 0.05, S0 = 0.2) competed by 20 μM A-395 with respect to 20 μM A-395N negative control. **c** Volcano plot of SGC2077-enriched proteome (labeled blue) from Jurkat cell lysate, with targets significantly (FDR = 0.05, S0 = 0.2) competed by 20 μM SGC0946, with respect to DMSO control. **d** STRING network evaluation of targets significantly competed by MS023. Lines in the STRING evaluation represent evidenced interactions, with line thickness indicating confidence (high to low). **e** STRING network evaluation of targets significantly competed by A-395. The chemical structures of the chemical biology reagents, MTM7172, (A-395)-NH₂, and SGC2077, are shown in Supplementary Table 2. See also Supplementary Fig. 1

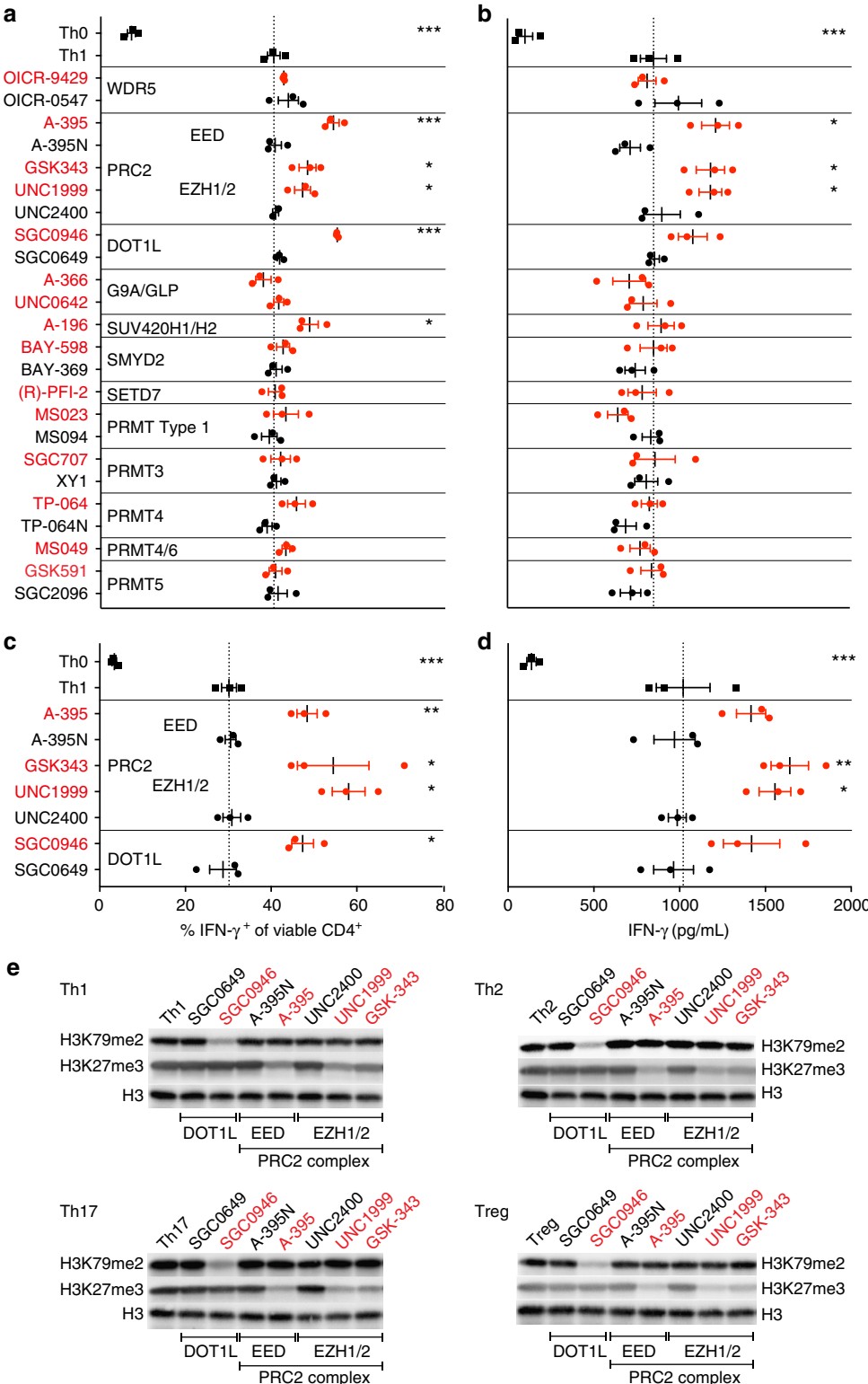

T cells using A-395 (but not A-395N), an antagonist of the EED subunit of PRC2 which prevents the latter's enzymatic activity[35], demonstrating that the enhanced IFN-γ expression and production was independent of the chemotype of the active probe, further strengthening the link between PRC2 inhibition and Th1 cell differentiation. Our results also identified a role for the H3K79 methyltransferase DOT1L in regulation of Th1 cell differentiation. DOT1L inhibition resulted in an increase in the frequency of viable IFN-γ⁺ CD4⁺ cells and higher production of

IFN-γ relative to the control compound and Th1 cells alone (Fig. 4a, b, and Supplementary Figs. 3a and 3b). To further confirm that these observed differentiation phenotypes were related to inhibition of the methyltransferase activity of PRC2 and DOT1L we performed western blot analyses of the histone methyl marks deposited by each enzyme. Th1 cells treated with UNC1999 or A-395 (but not with the control compounds UNC2400 or A-395N, respectively) showed a significant reduction of H3K27me3 relative to untreated cells. Similarly, western

**Fig. 4** Differential Effects of PMT Inhibition on Murine and Human Th1 Cell Differentiation. **a, b** CD4[+] T cells from the spleen and peripheral lymph nodes of IFN-γ-YFP reporter mice were enriched and polarized under Th0 (IL-2) or Th1 cell conditions in the absence (Th0) or presence of indicated probes (1 μM; red) or their controls (where available; black). **a** Flow cytometric analysis of intracellular YFP reporter signal (representing IFN-γ expression) was detected at day 4. **b** Secreted IFN-γ was analyzed by ELISA in the supernatant of the same experiment. Each data point represents one of three biological replicates and the data shown is representative of three independent experiments. **c, d** CD4[+] T cells from the blood of three healthy human donors were cultured under Th0 or Th1 cell-polarizing conditions in the presence or absence of indicated probes or their controls. **c** Flow cytometric analysis of intracellular IFN-γ was detected at day 4. **d** Secreted IFN-γ was analyzed by ELISA in the supernatant of the same experiment. Each data point represents one of three donors. Dotted lines visualize the mean frequency of IFN-γ-positive Th1 cells in the absence of probes (**a, c**) or the mean concentration of IFN-γ in the supernatant (**b, d**). Significant differences are indicated with an asterisk and were calculated using one-way ANOVA (*$p \le 0.05$, **$p \le 0.01$, ***$p \le 0.001$). **e** Western blot analysis of the effect of indicated inhibitors (red) or control compounds (black) on the trimethylation of H3K27 and dimethylation of H3K79 in CD4[+] T cells under Th1, Th2, Th17, Treg cell-polarizing conditions. Please see Supplementary Figs. 5a-5d for information on the MW markers. Data shown is representative of 2 independent experiments. Error bars represent SEM. See also Supplementary Figs. 2-5

blot analysis of H3K79me2 in the presence of SGC0946 showed an almost complete global loss of H3K79me2, but showed no change with the inactive SGC0649 or with the inhibitors of the PRC2 complex (A-395 or UNC1999) (Fig. 4e and Supplementary Fig. 5). Importantly, this specific inhibition was consistent across Th1, Th2, Th17, and Treg cells. Thus, inhibition of specific histone methyltransferases and subsequent loss of their specific histone methylation marks was associated with phenotypic changes in mouse Th cells.

**PMT regulation of Th1 cell responses is conserved in humans**. To determine whether human T cells responded in a manner similar to mouse T cells, we activated human naïve peripheral blood CD4[+] T cells for 4 days in the presence of Th1 cell-polarizing conditions. Consistent with our results for murine Th cells, inhibition of PRC2 (UNC1999, GSK343, A-395) and DOT1L (SGC0946) potentiated the effects of Th1 cell activation, resulting in a higher frequency of IFN-γ-producing Th1 cells as well as increased IFN-γ production, while the controls had no effect (Fig. 4c, d, and Supplementary Fig. 4). These results identify a central role for PRC2 and DOT1L in limiting Th1 cell differentiation in mice and humans.

Since to our knowledge DOT1L has not been examined in Th cell differentiation and function, we further investigated this enzyme using our mouse reporter system. In this four-day polarization assay, we only observed an enhancement of IFN-γ production with DOT1L inhibition if the cells were treated starting from day 0 or 1 of the culture, but not if the cells were exposed to the probe for only the last 1 or 2 days of polarization (Supplementary Fig. 3c). This is consistent with literature showing that reduction of histone methyl marks is dependent on the duration of exposure to methyltransferase inhibitors, often requiring days of exposure[49,50]. To explore the dynamics of H3K79me2 during Th1 cell differentiation, we monitored the reduction of H3K79me2 and the production of IFN-γ over a period of 4 days (Fig. 5). Western blot analysis of global H3K79me2 shows that the mark is reduced in Th cells following activation in the presence of the control compound SGC0649 (Fig. 5a), suggesting that Th cell activation alone leads to a reduction of H3K79me2 independent of DOT1L inhibition. However, inhibition of DOT1L by addition of SGC0946 resulted in further reduction of H3K79me2 by day 2 that was maintained until day 4 post activation (Fig. 5a). This earlier and increased reduction of H3K79me2 induced by SGC0946 correlated with heightened IFN-γ levels (Fig. 5b). These data establish a correlation between reduction of the H3K79me2 mark and enhanced production of IFN-γ and validate that reduced H3K79me2 correlates with increased production of IFN-γ under Th1 cell-polarizing conditions. To assess whether DOT1L inhibition affected T cell proliferation, we used flow cytometric tracking of naïve Th cells labeled with a fluorescent dye (CFSE),

and stimulated under either neutral (Th0) or Th1 cell-polarizing conditions in the presence of SGC0946 or SGC0649. We observed no effect on T cell proliferation (Supplementary Fig. 3d), suggesting that DOT1L inhibition likely affects the Th1 cell differentiation program without altering proliferative capacity.

We next carried out unbiased gene expression analysis. Naïve Th cells from IFN-γ-YFP mice were stimulated under Th1 cell-polarizing conditions for 4 days in the presence of SGC0946 or SGC0649. IFN-γ[+] CD4[+] Th1 cells were purified by cell sorting, and RNA was isolated for RNA-Seq analysis. Comparing SGC0946- and SGC0649-treated IFN-γ-positive Th1 cells, we observed 750 genes that were significantly upregulated, with 208 genes downregulated when DOT1L was inhibited (Fig. 5c). We observed that inhibition of DOT1L led to expression of non-canonical genes for Th1 cells including perforin 1 (*Prf1*), α7 integrin subunit (*Itga7*), and Ly6G (*Ly6g*). Thus, our data suggest that DOT1L-dependent mechanisms are potentially important for limiting Th1 cell differentiation and maintaining lineage integrity.

**PMTs differentially regulate Th cell differentiation**. We next extended our analysis of the probe collection to identify PMTs that may be involved in differentiation of not only Th1, but also Th2, Th17 or Treg cells. We activated naïve Th cells from either wild-type C57BL/6 mice or transgenic reporter mice (IFN-γ-YFP, IL-4-GFP, or FOXP3-EGFP) under Th1, Th2, Th17 or Treg cell differentiation conditions for four days in the absence or presence of PMT probes or control compounds, and analyzed expression of lineage-specific markers such as cytokines (Fig. 6) or transcription factors (Supplementary Fig. 7). As summarized in Fig. 6, we found a wide range of effects on T cell differentiation both between different probes as well as between different stimulation conditions for a single probe or chemotype. For example, inhibition of DOT1L and the PRC2 complex (EZH1/2, EED) enhanced effector cell differentiation (Th1, Th2, and Th17) with no effect on Treg cell development, while inhibition of G9a/GLP (UNC0642, A-366), strongly affected Treg cell differentiation. These data are consistent with our previous results demonstrating a key role for G9a in promoting Treg cell function[43,44].

Another chemical probe that showed strong promotion of Treg cell differentiation is MS023 which inhibits the type I PRMT enzymes, PRMT1, 3, 4, 6, and 8. PRMT8 is a neuronal-specific enzyme and not expressed in T cells[51]. MS023 was not tested on PRMT2 because the latter has not yet been demonstrated to be an active methyltransferase enzyme[52]. Comparison of this result with that of the more selective probes for PRMT3 (SGC707), PRMT4 (TP-064) and PRMT4/6 (MS049) all of which had little or no effect on Treg differentiation, suggests that the primary effect of MS023 is due to inhibition of PRMT1. This screen suggests potential strategies for promotion of Treg cell differentiation through inhibition of type I PRMTs, or PRMT1, should

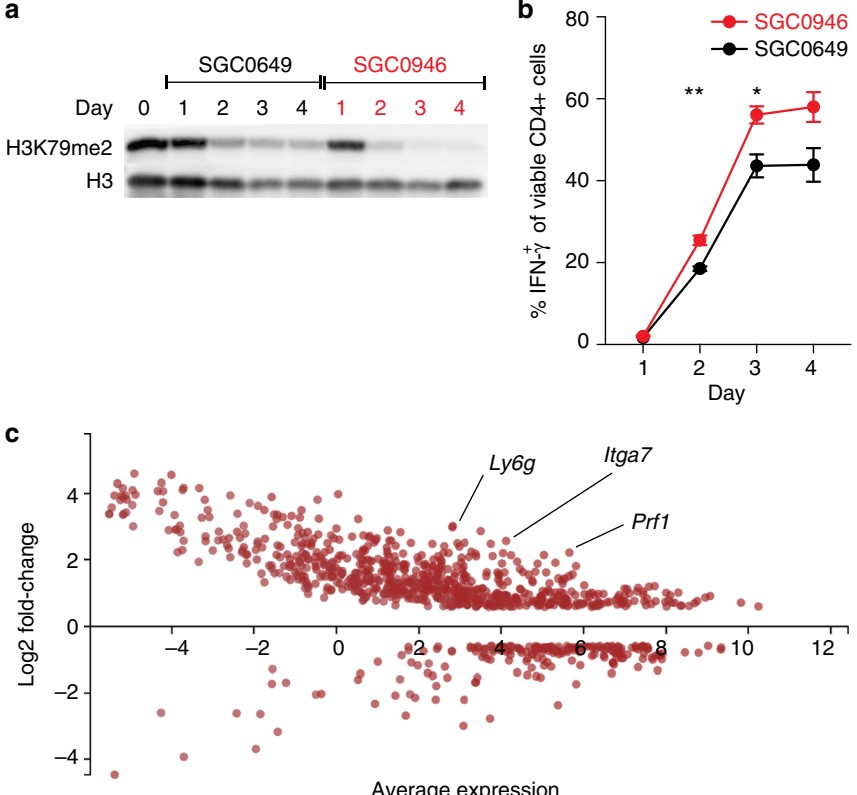

**Fig. 5** DOT1L-dependent H3K79me2 is dynamically regulated at lineage-specific promoters. **a** Analysis of H3K79me2 in CD4+ T cells under Th1 polarizing conditions in the presence of the DOT1L inhibitor SGC0946 (1 μM) or the control compound SGC0649 (1 μM) with time as measured by western blot. **b** Analysis of IFN-γ production by FACS under the same conditions as in **a**. Significant differences are indicated with an asterisk and were calculated using one-way ANOVA (*$p \leq 0.05$, **$p \leq 0.01$). Data shown are representative of three independent experiments. **c** DOT1L-dependent regulation of gene expression in Th1 cells. MA-plot (M (log ratio); A (mean average)) comparing SGC0649-treated and SGC0946-treated IFN-γ+ CD4+ T cells after Th1 polarization of up and down genes. Dots represent genes that show >1.5-fold increase (FDR 0.01) or decrease in SGC0946-treated CD4+ T cells compared to SGC0649-treated CD4+ T cells. Average expression is the mean of three biological replicates. Error bars represent SEM. See also Supplementary Figs. 3 and 5

a selective PRMT1 probe become available. Such compounds may be beneficial to subdue aberrant inflammatory immune responses.

Inhibition of PRMT5 using peptide competitive (GSK591) and SAM competitive (LLY-283) chemical probes clearly downregulated the signature cytokines for effector cell differentiation, but also significantly increased cell death (Supplementary Fig. 7). PRMT5 has been shown to be required for growth and viability of rapidly proliferating cells such as cancer cell lines[53]. Although the mechanisms are not fully understood, PRMT5 knockdown slows the cell cycle in NIH3T3 cells and induces G1 arrest in 293T and MCF7 cells[54,55]. It is possible that similar mechanisms are at play in proliferating Th cells. The effect of PRMT5 on Treg cell differentiation is unclear since the chemical probes yielded opposite results (even across 6 biological replicates).

## Discussion
Here we present a collection of chemical biology reagents to modulate cellular methylation signaling, especially chromatin-mediated signaling. Each probe has been characterized for its selectivity within the human SET and PRMT methyltransferase families. Most of our chemical probes are accompanied by structurally similar inactive compounds that serve as negative controls for potential off-target effects of their common chemical scaffold. We have annotated the chemical structure of each chemical probe showing where they can be chemically derivatized to create additional reagents such as biotinylated probes for chemo-

proteomics. Using several examples of such derivatives we demonstrate the cellular selectivity for their targets and respective interacting proteins. Importantly, these probes and related compounds may be used for research without restrictions.

Using our collection of PMT chemical probes, we identified several PMTs that differentially regulate Th cell differentiation. For example, inhibition of the PRC2 complex led to enhanced Th1, Th2 and Th17 cell responses and a reduction in Treg cell development, which is consistent with the phenotype observed in mice with a genetic deletion of EZH2[45]. In addition, inhibition of G9a resulted in a significant increase in the frequency of FOXP3-expressing Treg cells, which is in agreement with our previous results[44]. Our unbiased screen also identified regulators of Th cell differentiation. Inhibition of DOT1L promoted effector Th cell responses with little effect on Treg cell differentiation, as did type I PRMT inhibition. While the potential enhancement of a regulatory immune response by inhibitors of G9a or by members of the Type I PRMT family may be beneficial to control immune responses in inflammatory disease, an increase of pro-inflammatory responses by inhibiting the PRC2 complex or DOT1L might be beneficial to boost the immune response in underperforming immune systems. These results warrant further investigation given the intense interest in T cell biology and the growing appreciation for the role of epigenetics in T cell differentiation[56–58]. Thus, the PMT chemical probe library provides a toolbox to examine the biological role of methylation-mediated signaling in cellular assays.

| | Complex | Target | Chemical probe | Th1 | fc | Th2 | fc | Th17 | fc | Treg | fc |
|---|---|---|---|---|---|---|---|---|---|---|---|
| **Protein lysine methyltransferases (PKMTs)** | MLL1 | WDR5 | OICR-9429 | | 0.9 | | 0.8 | | 1.2 | | 1.0 |
| | PRC2 | EED | A-395 | | 1.3 | | 1.1 | | 1.5 | | 1.0 |
| | | EZH2/H1 | GSK343 | | 1.1 | | 1.0 | | 1.1 | | 0.9 |
| | | | UNC1999 | | 1.2 | | 1.4 | | 1.3 | | 0.9 |
| | DOT | DOT1L | SGC0946 | | 1.5 | | 1.1 | | 2.4 | | 1.0 |
| | G9a/GLP | G9a,GLP | A-366 | | 0.8 | | 1.0 | | 1.4 | | 1.4 |
| | | | UNC0642 | | 0.9 | | 1.2 | | 1.4 | | 1.4 |
| | | SUV420H1/H2 | A-196 | | 1.3 | | 1.1 | | 2.3 | | 0.9 |
| | | SMYD2 | BAY-598 | | 0.9 | | 0.8 | | 1.1 | | 1.1 |
| | | SETD7 | PFI-2 | | 0.9 | | 1.0 | | 1.0 | | 1.1 |
| **Protein arginine methyltransferases (PRMTs)** | | PRMT type I | MS023 | | 0.9 | | 1.5 | | 1.0 | | 1.5 |
| | | PRMT3 | SGC707 | | 1.0 | | 0.8 | | 1.0 | | 1.0 |
| | | PRMT4 | TP-064 | | 1.1 | | 1.1 | | 1.0 | | 1.2 |
| | | PRMT5 | GSK591 | | * 0.3 | | * 0.1 | | * 0.2 | | 1.2 |
| | | | LLY-283 | | * 0.2 | | * 0.1 | | * 0.1 | | * 0.8 |
| | | PRMT4/6 | MS049 | | 1.0 | | 1.1 | | 1.0 | | 1.0 |

fc - fold change
* - significant cell death observed

**Fig. 6** Overview of PMT Inhibition on Murine CD4[+] T cell Differentiation. Data is presented as fold-increase (black) or decrease (red) of cytokine production (IFN-γ, IL-13, and IL-17A for Th1, Th2, and Th17 cells, respectively) or FOXP3 expression (Treg cells) over that of the control compound-treated (or untreated) cells. Data shown are from two to three individual experiments with three biological replicates each. *Significant cell death: *$p < 0.05$ compared to DMSO control. See also Supplementary Figs. 6 and 7

One striking finding from our results is that specific probes displayed diverse functional effects on distinct Th cell subsets despite global reductions in their respective specific histone modification in all subsets examined. For example, inhibition of DOT1L led to heightened effector T cell responses with minimal effects on Treg cells, while inhibition of G9a primarily affected Th17 and Treg cells. Although the precise molecular mechanisms are yet to be elucidated, these results suggest that the targeted epigenetic modifiers do not act in isolation but rather collaborate with lineage-specific factors that are expressed in the specific Th cell subsets. As the PMTs do not have sequence-specific DNA-binding domains, Th cell subset-specific accessory factors that target PMTs to the chromatin will be critical modulators of function. In addition, it is possible that non-histone substrates of PMTs are critically important in some Th cell subsets. Thus, this chemical probe library provides tools to examine the fundamental mechanisms associated with epigenetic regulation of Th cells.

Among the targets in our chemical probe collection, several have agents currently in clinical trials for cancer (EZH2, EED, DOT1L, PRMT1, and PRMT5) and additional PMT inhibitors are in preclinical studies. The availability of well characterized, unencumbered chemical probes for such targets will enable the research community to better understand the mechanisms and consequences of PMT inhibition in a wide variety of cellular contexts providing important knowledge to help guide clinical development, patient stratification, and novel applications of PMT inhibitors. Of particular interest are the apparent pro-inflammatory activities of PRC2 and DOT1L inhibitors, which warrant further investigation in the immune-oncology setting.

Overall, our collection of 19 chemical probes to 16 PMT targets, 17 chemotype-matched controls, and 18 related chemical biology reagents will be an outstanding resource to enable research in methylation-dependent epigenetic regulation. Our detailed descriptions of the potency, selectivity and structural mechanisms of inhibition of each chemical probe provide a comprehensive overview of this target class and should facilitate the development of new chemical probes to other PMTs. Additional uses of the collection may include (i) synthetic lethal screens of the probes versus genetic ablation of individual genes in cancer, (ii) screens for optimal combinations of PMT probes with existing standard of care therapies in cellular disease models, (iii) investigation of endogenous PMT protein complexes in a variety of cells without the need to introduce tagged exogenous protein baits, and (iv) development of protein degrading reagents through derivatization of chemical probes[59].

## Methods

**Methyltransferase selectivity assays.** The effects of chemical probes and negative controls on the methyltransferase activities of protein, DNA and RNA methyltransferases were tested by radiometric assays using [3]H-SAM. For proteins such as MLL1 trimeric complex, MLL3 pentameric complex, EZH1 (PRC2) pentameric complex, EZH2 (PRC2) trimeric complex, as well as G9a, GLP, SUV39H1, SUV39H2, SUV420H1, SUV420H2, SETD2, SETD8, SETDB1, SETD7, PRMT1, PRMT3, PRMT4, PRMT5/ MEP50 complex, PRMT6, PRMT7, PRMT8, PRMT9, PRDM9, SMYD2, SMYD3, DNMT1 and BCDIN3D the incorporation of a tritium-labeled methyl group into biotinylated substrate (Supplementary Table 3) was monitored using scintillation proximity assay (SPA). Briefly, a 10-μL reaction containing [3]H-SAM and substrate at concentrations close to the apparent Km values for each enzyme (balanced conditions) was prepared. The reactions were quenched with 10 μL of 7.5 M guanidine hydrochloride; 180 μL of 20 mM Tris buffer (pH 8.0) were added, and the mixture was transferred to a 96-well FlashPlate and incubated for 1 h. The counts per minute (CPM) was measured on a TopCount plate reader. The CPM in the absence of compound or enzyme was defined as 100% activity and background (0%), respectively, for each dataset.

For DNMT1, the double-stranded DNA substrate was prepared by annealing two complementary strands (biotinylated forward strand: B-GAGCCCGTAAGC CGGTTCAGGTCG and reverse strand: CGACCTGAACGGGCTTACGGGCTC) that were synthesized by Eurofins MWG Operon (Louisville, KY, USA).

For proteins which were tested with nucleosome as substrate such as DOT1L, NSD1, NSD2, NSD3 and ASH1L, or unbiotinylated Poly(2′-deoxyinosinic-2′-deoxycytidylic acid) (Cat# 81349-500UG, Sigma Aldrich) such as DNMT3A/3L, and DNMT3B/3L, a filter-based assay was used. In this assay, a trichloroacetic acid (TCA) protein precipitation protocol was employed. A 10 μL reaction mixture was incubated at 23 °C for 1 h, followed by addition of 50 μL of 10% TCA. The mixture was transferred to filter plates (Millipore, Billerica, MA, USA) that were centrifuged at 931 × g (Allegra X-15R; Beckman Coulter, Brea, CA, USA) for 2 min. Samples were washed twice with 10% TCA and once with ethanol (180 μL), and centrifuged (as before). After drying, 100 μL MicroScint-O (Perkin Elmer) was added to each well and the plates were centrifuged to remove the liquid. A 70-μL volume of MicroScint-O was added and the CPM was measured with a TopCount plate reader.

**Mice.** C57BL/6 mice and reporter mice for IFN-γ[47], IL-4[60], or Foxp3[61] on C57BL/6 background were used for the polarization assays, where possible. Where unavailable, intracellular staining for cytokines and transcription factors was performed. All animal experiments were approved by the Monash University Animal Care Committee.

**T cell polarization, proliferation and flow cytometry.** Mouse CD4[+] T cells were isolated with the CD4[+] T Cell-Negative Isolation Kit (Stemcell Technologies) and polarized for optimal results for 4 days under Th0, Th1, Th2, Th17 and Treg conditions[44]. Here, 175,000 naïve CD4[+] T cells were cultured under Th0 (IL-2 [10 ng mL$^{-1}$]), Th1 (IL-2, IL-12 [10 ng mL$^{-1}$ each], anti-IL-4 [10 μg mL$^{-1}$]), Th2 (IL-2, IL-4 [10 ng mL$^{-1}$ each], anti IFN-γ [10 μg mL$^{-1}$]), Th17 (IL-23, IL-1β, TNF-α [10 ng mL$^{-1}$ each], IL-6 [20 ng mL$^{-1}$], TGF-β [1 ng mL$^{-1}$], anti-IL-4 and anti-IFN-γ [10 μg mL$^{-1}$ each] or Treg conditions (IL-2 and TGF-β [10 ng mL$^{-1}$ each] in 96-well plates (pre-coated overnight with 1 μg mL$^{-1}$ of each anti-CD3 and anti-CD28 in PBS) in 200 μL complete RPMI media (10% FCS, 2 mM L-glutamine, 100 U mL$^{-1}$ penicillin, 100 μg mL$^{-1}$ streptomycin, 25 mM HEPES, 50 μM β-mercaptoethanol) in the absence or presence of indicated amounts of the chemical probes. Viability of the cells was determined using fixable viability dye (ThermoFisher Cat# 65-0863-18; 1/1000). Signature cytokines and transcription factors were detected using reporter mice or by intracellular staining 4 h after incubation with the cytokine stimulation cocktail (ThermoFisher Cat# 00-4975-03; 1/500) and antibodies against TBET (eBio4B10; ThermoFisher Cat# 25-5825-82; 1/200), GATA3 (TWAJ; ThermoFisher Cat# 12-9966-42; 1/200), RORγt (B2D; ThermoFisher Cat# 17-6981-82; 1/200), Foxp3 (FJK-16S; ThermoFisher Cat# 17-5773-82), IFN-γ (XMG1.2; ThermoFisher Cat# 25-7311-82; 1/200), IL-13 (eBio13A; ThermoFisher Cat# 12-7133-82; 1/200) or IL-17 (eBio17B7; Thermo-Fisher Cat#12-7177-81; 1/200) using the intracellular fixation & permeabilization buffer set (ThermoFisher Cat# 88-8824-00) or the Foxp3/Transcription factor staining buffer set (ThermoFisher Cat# 00-5523-00) according to the manufacturer's inductions. Proliferation assay of CFSE labeled mouse CD4[+] T cells was performed under Th0 or Th1 polarizing conditions and analyzed at day 3.

Naïve CD4[+] T cells were isolated from blood samples of three healthy donors using the human naïve CD4[+] T Cell Isolation Kit II (Milteny). One hundred twenty thousand naïve CD4[+] T cells were polarized towards Th1 cells using anti-CD3/CD28 antibodies (1 bead:5 cells ratio) in the presence of recombinant IL-12 [10 ng mL$^{-1}$], anti-IL4 antibody [10 mg mL$^{-1}$] and recombinant IL-2 [10 ng mL$^{-1}$] for 4 days in the presence of chemical probes or their respective probe controls at a concentration of 1 μM which is in the range of the cellular IC$_{50}$-IC$_{90}$ of most of the active chemical probes. The media was not changed and compounds were not replenished over the duration of the experiment. Naïve CD4[+] T cells were maintained in culture in the presence of IL-2 [10 ng mL$^{-1}$] for 4 days and used as a control. Cells were stained with SYTOX™ Blue Dead Cell Stain (Thermo Fisher Scientific), fixed and permeabilized, followed by the intracellular staining with anti-IFN-γ antibody. All healthy volunteers accepted to donate their blood samples for research purposes by signing an informed consent (Mount Sinai Hospital, Research Ethics Board #02-0234-E).

**RNA-Seq and bioinformatics.** Naïve CD4[+] T cells (CD44neg (Thermo Fisher Scientific Cat# 48-0441-80; 1/200), CD62Lhi (Thermo Fisher Scientific Cat# 25-0621-82; 1/200)) and CD4[+] T cells positive (IFN-γ[+]) or negative (IFN-γ[−]) for YFP after Th1 polarization were sorted and total RNA was extracted using the Nucleospin RNA kit (Macherey-Nagel), according to the manufacturer's instructions. RNA was isolated with an mRNA kit (TruSeq Stranded; Illumina, San Diego, CA, USA) and sequenced on a MiSeq paired-end run (75 × 75, v3; Illumina). Samples were aligned to the mm10 transcript reference using TopHat2, and differential expression was assessed using Cufflinks (Illumina). Visualization of the data was performed using DEGUST (https://github.com/drpowell/degust) and represent the average expression from 3 biological replicates (x-axis) and the Log2-fold change of SGC0946-treated cells over SGC0649-treated cells (y-axis).

**Western blot assays.** Enriched CD4[+] T cells were cultured in the absence (Th1) or presence of indicated chemical probes for 4 days (or as indicated in the experiment) under Th1 polarizing conditions. Cells were harvested and pellets were frozen at −80 °C. Histones were extracted from frozen cell pellets by incubating in 0.2 N HCl overnight at 4 °C. Supernatants were run on 12% SDS-PAGE gels. H3K27me3 and H3K79me2 were detected using clones ab6002 and ab3594 (Abcam), respectively. A pan anti-histone H3 antibody (ab1791, Abcam) at a concentration of 1 μg mL$^{-1}$ was used as a loading control. The uncropped blots are shown in Supplementary Fig. 5.

**Chemical proteomics.** G401, Jurkat and HEK293T were obtained from ATCC (Virginia, USA) and cultured at 37 °C in a humidified 5% CO2 atmosphere in McCoy's Medium containing 10% FBS, RPMI-1640 containing 10% FBS, and DMEM Medium containing 10% FBS, respectively. The cell lines are not found in the ICLAC database for commonly misidentified cell lines, and were not authenticated. All cell lines tested negative for mycoplasma contamination (MycoAlert™ PLUS Mycoplasma Detection Kit, Lonza). For cell lysate experiments, cells were grown until approximately 80% confluency before being pelleted and washed with

PBS. Cell pellets were subsequently lysed by addition of Buffer A (50 mM Tris pH 7.5, 0.8% v/v NP-40, 5% v/v glycerol, 1.5 mM MgCl$_2$, 100 mM NaCl, 25 mM NaF, 1 mM Na$_3$VO$_4$, 1 mM PMSF, 1 mM DTT, 10 μg mL$^{-1}$ TLCK, 1 μg mL$^{-1}$ Leupeptin, 1 μg mL$^{-1}$ Aprotinin, 1 μg mL$^{-1}$ soy bean trypsin) on ice, followed by ten passes through a 21G needle. Following 30 min incubation on ice, crude lysates were cleared by ultracentrifugation (86,900 × g, 4 °C, 1 h), protein concentration was determined and lysates stored at −80 °C until use[14].

Eluted proteins were separated on polyacrylamide gels with SDS running buffer (50 mM MES, 50 mM Tris Base, 0.1% SDS, 1 mM EDTA, pH 7.3) and transferred to nitrocellulose blotting membranes. Membranes were blocked with blocking buffer (2.5% (m/v) BLOT-QuickBlocker (Merck) in PBST (Phosphate-buffered saline with Tween: 4.3 mM Na$_2$HPO$_4$, 1.47 mM KH$_2$PO$_4$, 137 mM NaCl, 2.7 mM KCl, 0.05% (v/v) Tween 20) before probing with the antibodies mouse anti-EED (clone GT671; Thermo Fisher Scientific Cat# MA5-16314; 1:1,000), and CARM1 (Bethyl Laboratories Inc. Cat# A300-421A; 1:10,000). The uncropped blots are shown in Supplementary Fig. 1.

Amine derivatized compounds were coupled to NHS-activated Sepharose 4 fast flow beads (GE Healthcare)[36]. 100 μL of bead slurry (50% in isopropanol) were used for each pull-down experiment. Beads were washed with DMSO (500 μL), collected by centrifugation (60 × g, 3 min), and the supernatant removed. After three wash cycles, the beads were re-suspended in DMSO (50 μL), to which the amine (0.025 μmol) and triethylamine (0.75 μL) were added. The beads were incubated at room temperature for 16 h, and depletion of free amine from the supernatant determined by LC-MS analysis. Ethanolamine (2.5 μL) was then added to block any unreacted NHS sites, and the beads incubated for a further 16 h. Derivatized beads were then washed with DMSO (3 × 500 μL), Buffer A (3 × 1 mL), and incubated with cell lysates (2 mg of protein per pulldown, at 6 mg mL$^{-1}$) that had been pre-treated with either compound (20 μM) or DMSO control for 30 min at 4 °C. Beads and treated lysates were incubated for 2 h at 4 °C, before being washed with Buffer A (5 mL), Buffer B (50 mM HEPES pH 7.5, 100 mM NaCl, 500 μM EDTA, 2.5 mL), and eluted with formic acid (100 mM, 250 μL). Samples were neutralized with triethylammoniumibicarbonate (TEAB, 1 M, 62.5 μL) and stored at −20 °C until preparation for proteomic analysis.

Biotin derivatized compounds were coupled to UltraLink Immobilized Streptavidin Plus beads (GE Healthcare)[6]. 100 μL of bead slurry (50% in isopropanol) were used for each pulldown experiment. Beads were washed with Buffer A (500 μL), collected by centrifugation (60 × g, 3 min), and the supernatant removed. After three wash cycles the beads were re-suspended in Buffer A (1 mL), to which the biotinylated compound was added (0.05 μmol) and incubated for 30 min at 4 °C followed by a final wash step with Buffer A (2 × 1 mL). Lysates were precleared by the addition of 100 μL of bead slurry (50% in isopropanol) and incubated for 30 min at 4 °C. After preclearing, lysates were treated with either compound at the indicated concentration or DMSO control for 30 min at 4 °C followed by incubation with the affinity matrices for two hours at 4 °C. Affinity matrices were washed with buffer A (5 mL), Buffer B (50 mM HEPES pH 7.5, 100 mM NaCl, 500 μM EDTA, 2.5 mL), and proteins eluted with Buffer C (3 M Urea, 50 mM formic acid, 10 mM DTT, 250 μL). Samples were neutralized with triethylammonium bicarbonate (TEAB, 100 mM, 30 μL) and stored at −20 °C until preparation for proteomic analysis. For Western blot experiments bound proteins were eluted by addition of 100 μL of 2× sample buffer (65.8 mM Tris-HCl pH 6.8, 26.3% (w/v) glycerol, 2.1% SDS, 0.01% bromophenol blue, 50 mM DTT).

Samples were reduced with DTT (10 mM final concentration) for 30 min at room temperature, alkylated with iodoacetamide (55 mM final concentration) for 30 min at room temperature, diluted to 300 μL with TEAB, and incubated with trypsin (6 μL, 0.2 mg mL$^{-1}$) overnight at 37 °C. The digests were then desalted using SEPAC lite columns (Waters), eluted with 69% v/v MeCN, 0.1% v/v FA in H$_2$O (1 mL) and dried in vacuo. Dried peptides were stored at −20 °C before resuspension in 2% v/v MeCN, 0.1% v/v FA in H$_2$O (20 μL) for LC-MS/MS analysis

Mass spectrometry data were acquired at the Discovery Proteomics Facility (University of Oxford). Digested samples were analyzed by nano-UPLC–MS/MS using a Dionex Ultimate 3000 nano UPLC with EASY spray column (75 μm × 500 mm, 2 μm particle size, Thermo Scientific) with a 60 min gradient of 0.1% (v/v) formic acid in 5% (v/v) DMSO to 0.1% (v/v) formic acid with 35% (v/v) acetonitrile in 5% (v/v) DMSO at a flow rate of approximately 250 nL min$^{-1}$ (600 bar per 40 °C column temperature). Mass spectrometry data were acquired either with an Orbitrap Q Exactive (survey scans acquired at a resolution of 70,000 @ 200 m/z and the 15 most abundant precursors were selected for HCD fragmentation), or an Orbitrap Q Exactive High Field (HF) instrument (survey scans were acquired at a resolution of 60,000 at 400 m/z and the 20 most abundant precursors were selected for CID fragmentation).

Raw data was processed using MaxQuant version 1.5.0.253 and the reference complete human proteome FASTA file (UniProt). Label Free Quantification (LFQ) and Match Between Runs were selected; replicates were collated into parameter groups to ensure matching between replicates only. Cysteine carbamidomethylation was selected as a fixed modification, and methionine oxidation as a variable modification. Default settings for identification and quantification were used. Specifically, a minimum peptide length of 7, a maximum of 2 missed cleavage sites, and a maximum of 3 labeled amino acids per peptide were employed. Through selection of the 'trypsin/P' general setting, peptide bond cleavage at arginine or lysine (followed by any amino acid) was considered during in silico digest of the reference proteome. The allowed precursor and fragment ion mass tolerances were

4.5 ppm and 20 ppm, respectively. Peptides and proteins were identified utilizing a 0.01 false discovery rate, with "Unique and razor peptides" mode selected for both identification and quantification of proteins (razor peptides are uniquely assigned to protein groups and not to individual proteins). At least 2 razor + unique peptides were required for valid quantification. Processed data was further analyzed using Perseus version 1.5.0.9 and Microsoft Excel 2010. Peptides categorized by MaxQuant as 'potential contaminants', 'only identified by site' or 'reverse' were filtered, and the LFQ intensities transformed by log2. Experimental replicates were grouped, and two valid LFQ values were required in at least one experimental group. Missing values were imputed using default settings, and the data distribution visually inspected to ensure that a normal distribution was maintained. Statistically significant competition was determined through the application of P2 tests, using a permutation-based FDR of 0.05 and an S0 of 2, and visualized in volcano plots. Significantly competed targets were further analyzed in STRING (http://string-db.org) and protein interaction networks generated. Basic STRING settings were used for network analysis of enriched proteins. Specifically, network edges represent confidence in interaction. Line thickness indicates the strength of data support with a minimum required interaction score of 0.400. All active interaction sources (Textmining, Experiments, Databases, Co-expression, Neighborhood, Gene Fusion, Co-occurrence) were considered.

**Syntheses of reagents**. The syntheses of the reagents are described in the Supplementary Information.

**Reporting summary**. Further information on experimental design is available in the Nature Research Reporting Summary linked to this article.

## Data availability

The proteomics data that support the findings of this study have been deposited in the PRIDE partner repository[62](ProteomeXchange Consortium) with the dataset identifier PXD009028. The RNA-Seq data that support the findings of this study are available at the National Center for Biotechnology Information with the primary accession code GSE106978. All other data that support the findings of this study are available from the corresponding author on reasonable request. A reporting summary for this Article is available as a Supplementary Information file.

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

## Acknowledgements

We thank our numerous medicinal chemistry collaborators in the pharmaceutical industry, the Ontario Institute for Cancer Research (OICR), and the Jin, Frye, and Luo Labs who helped develop probes in this resource. Michael Curtin and Hongyu Zhou (Abbvie) contributed to the development of (A-395)-NH$_2$ and (A-395)-biotin. Stefan Gradl and Carlo Stresemann (Bayer) spearheaded the discovery of BAY-6035. Kazuhide Nakayama (Takeda) contributed to the development of TP-064. Hiroshi Nara and Nozomu Sakai (Takeda) contributed to the discovery of SGC3027. 7TM, kinase, and ion channel off-target selectivity screening was kindly supplied by Eurofins-Cerep. Additional Ki determinations and receptor binding profiles were generously provided by the National Institute of Mental Health's (NIMH) Psychoactive Drug Screening Program, (Contract # HHSN-271-2013-00017-C) directed by B.L. Roth, University of North Carolina at Chapel Hill and Project Officer J. Driscoll at NIMH, Bethesda, Maryland, USA. We thank B. Kessler, S. Bonham and R. Fischer from the TDI Discovery Proteomics Facility, Oxford University, for their support. We acknowledge Mark Silverberg, Michelle Smith, Beatrice Luu and Grace Chan from Mount Sinai Hospital (Canada) for their contribution in the collection of human blood samples used in this study. This work was supported by the Canadian Institutes of Health Research [FDN-154328 and 128090 to C.H.A., and FDN-148430 and 201512MSH-360794-228629 to D.D.C.], Canadian Cancer Society [CCSRI 703716 to D.D.C.], the Australian National Health and Medical Research Council (project grants 1104433 and 1104466 to C.Z.), Myeloma UK [J.A.W., A.M.L., and K.V.M.H.], the U.S. National Institutes of Health [R01GM122749, R01CA218600, and R01HD088626 to J.J.], the OICR Drug Discovery Program (funded by the government of Ontario), and the Structural Genomics Consortium (SGC) which is a registered charity (number 1097737) that receives funds from AbbVie, Bayer Pharma AG, Boehringer Ingelheim, Canada Foundation for Innovation, Eshelman Institute for Innovation, Genome Canada through Ontario Genomics Institute [OGI-055], Innovative Medicines Initiative (EU/EFPIA) [ULTRA-DD grant no. 115766], Janssen, Merck KGaA, Darmstadt, Germany, MSD, Novartis Pharma AG, Ontario Ministry of Research, Innovation and Science (MRIS), Pfizer, São Paulo Research Foundation-FAPESP, Takeda, and Wellcome. T.M. is supported by a fellowship "Conselho Nacional de Desenvolvimento Científico e Tecnológico (CNPq)", Brazil. C.Z. is a VESKI Innovation Fellow.

## Author contributions

S.S. performed murine T cell experiments. T.S.M. performed human T cell experiments. J.A.W. and A.M.L. performed proteomics experiments. J.P.N. performed western blots. F.L. performed vitro PMT assays. M.S. analyzed protein structures. D.S., D.M., H.U.K., Y.S., P.L.R., C.A.Z.-V. and J.L. synthesized compounds. M.L., C.Z., D.D.D.C., K.V.M.H., M.V., P.B., J.J., D.B.-L. and C.H.A. conceived and supervised research. M.V., S.S., T.S.M., M.S., P.J.B., C.Z., S.A., and C.H.A. wrote the manuscript. C.H.A. oversaw the project.

## Additional information

**Competing interests:** Paul L. Richardson is an employee of AbbVie and holds AbbVie stock. The remaining authors declare no competing interests.

