## [Peer Review File · Nature Communications]

Reviewer #1 (Remarks to the Author):

Scheer and colleagues report the description of an epigenetics probe compound library targeting 15 different chromatin regulators and their application in the context of T-cell lineages. A library of these well characterized chemical compound was utilized to explore whether any of the chromatin regulators would specifically affect the differentiation of naïve T-cells into Th1, Th2, Th17 or Treg lineages. Based on prior literature, it is clear that chromatin regulators are important players in governing T-cell differentiation and function, and the described chemical biology approach would allow to identify the most prominent regulators for each T-cell lineage in comparison to other chromatin regulatory mechanisms.

The experiments in this manuscript are generally well performed and yield clear data. Some of the preliminary findings from their 'T-cell screen' are potentially interesting, however, they lack the depth and the necessary follow up work required to qualify any of their 'hits' as potential target for immuno-inflammatory or immuno-oncology applications. Moreover, the first two figures fit better into a review article rather than into a primary research article since all these probes and their respective mechanisms of action have been described previously. Figure 3 describes a few new biotinylated derivatives of described tool compounds. These probes are potentially interesting in their own rights, however, for the purpose of this manuscript serve merely as a means to obtain additional selectivity data for probe compounds that were already well characterized with respect to their advertised selectivity from prior publications. Figures 4-6 are focused on identifying factors that impact Th1 cell differentiation and they confirm genetic data implicating EZH2/PRC2 in this process. A novel piece of data is provided by the finding that the DOT1L methyltransferase inhibitor promotes Th1 differentiation by increasing the percentage of IFN gamma-positive Th1 cells. While this is an original finding and while DOT1L inhibitor-treated Th1 cells experience hundreds of gene expression changes that distinguish them from control Th1 cells it remains unclear how inhibition of DOT1L affects Th1 biology. It also remains unclear whether other lymphocyte subsets are positively or negatively impacted by DOT1L inhibition, an assessment that is necessary to suggest an application as immune-modulating agent for human disease.

Overall, the manuscript cannot be recommended for publication in Nature Communications since it does not contain sufficient novel data or a coherent piece of data (e.g. around better defining the immune phenotypes upon DOT1L inhibition) that significantly advances our understanding of how an epigenetic drug could be utilized for immune modulation in human disease.

Reviewer #2 (Remarks to the Author):

In "A Chemical Biology Toolbox to Study Protein Methyltransferases and Epigenetic Signaling" the authors compile and validate potency and selectivity information on a well characterized set of methyltransferase probes. They then go on to use this set of 17 probes to rigorously screen for different cellular effects in T cell activation and differentiation. Overall, this paper represents an important resource for the community. In addition, the results with T cells are quite intriguing.

1. Comparative western blots (similar to what was done in Figure 4E) for the EED, EZH2/H1, DOT1L and G9a/GLP inhibitor results in the different T cell subsets would be helpful.

2. As well as the cytokine expression data already presented, it would be helpful to know if any of the T cell subsets from Figure 6 had actual enhanced (or decreased) function due to treatment with different inhibitors.

Reviewer #3 (Remarks to the Author):

The manuscript by Scheer et al. provide comprehensive summary for chemical probes that were developed by the SGC group against multiple of histone methyltransferases. While these chemical probes have been described in previous publications for different biological context, this work is the first comprehensive study for the collection of histone methyltransferase inhibitors on T cell activation and differentiation.

There are several technical concerns regarding the T cell studies:

1. Experiments in Figure 4 were confusing. The legend said "under Th1 cell condition in the absence (Th0) or presence of indicated probes (1 μ M; red) or their controls (where available; black)." Did all probes induce Th0 to Th1 differentiation while positive inhibitors induces IFN- γ ? Or all cells used here were actually Th1 cells?

2. Expression of signature T cell genes, e.g. T-bet for Th1, GATA3 for Th2 etc. should be provided in addition to cytokine induction.

3. Figure 6 was also confusing. What was the control? What was the reference for fold-change? Was it treatment by negative control compounds? Showing means \pm S.D might be better way to present the data.

4. The study on T cell differentiation was descriptive. It would be better to include some mechanistic insights on why DOT1L affect Th1 and Th17 specifically while G9a affects mostly T-reg etc. Expansion on this point, especially the role of DOT1L in T-cell differentiation, will greatly increase the significance of this study.

5. It would be very useful to include information on whether these probes could be used in vivo.

Reviewers' comments:

Reviewer #1 (Remarks to the Author):

Scheer and colleagues report the description of an epigenetics probe compound library targeting 15 different chromatin regulators and their application in the context of T-cell lineages. A library of these well characterized chemical compound was utilized to explore whether any of the chromatin regulators would specifically affect the differentiation of naïve T-cells into Th1, Th2, Th17 or Treg lineages. Based on prior literature, it is clear that chromatin regulators are important players in governing T-cell differentiation and function, and the described chemical biology approach would allow to identify the most prominent regulators for each T-cell lineage in comparison to other chromatin regulatory mechanisms.

1a. The experiments in this manuscript are generally well performed and yield clear data.

We are glad the reviewer is confident in the quality of our data.

1b. Some of the preliminary findings from their 'T-cell screen' are potentially interesting, however, they lack the depth and the necessary follow up work required to qualify any of their 'hits' as potential target for immuno-inflammatory or immuno-oncology applications.

Our T-cell screen is meant to identify epigenetic targets whose pharmacologic inhibition could be potentially interesting for understanding or treating human disease. As such, we believe that although preliminary, this dataset holds a wealth of starting points for further study of T-cell mediated biology or disease. Furthermore, because several of the targets (EZH2 and G9a) have published genetic data that phenocopy our screening results, we believe there is good reason to expect that the remaining "hits" will have interesting biology associated with them.

We would like to point out that a key rationale for the T cell differentiation screen was to highlight, (a) how one can use the probe collection (including appropriate use of inactive controls) and (b) how each chemical probe can have a unique combination of phenotypes across the T cell lineages, despite universally depleting their respective mark in each lineage. This latter point is important because with our current state of understanding of epigenetics we are not able to predict *a priori* how a given cell type will respond to inhibition of a given epigenetic regulator. Discovery of new biology resulting from modulation of epigenetic regulators will need to be done in an empirical manner, at least for the time being. Thus, our probes will be essential for such studies.

Nevertheless, we agree with the reviewer that much further work will be required to validate any of these targets for inflammatory disease or immuno-oncology applications. As such, we have 'toned down' the wording in our abstract to "We demonstrate the utility of this collection in CD4⁺ T cell differentiation assays revealing the remarkable potential of individual probes to alter multiple T cell subpopulations with **potential** implications for T cell-mediated processes such as inflammation and immuno-oncology."

1c. Moreover, the first two figures fit better into a review article rather than into a primary research article since all these probes and their respective mechanisms of action have been described previously.

Yes, there are publications for most of these probes, however, in our revised manuscript we report two new probes (BAY-6035 and SGC3027) in addition to SKI-73, all three of which have not been published before. We believe these are important new data. Furthermore, the heatmap of biochemical selectivity (Fig 1C) comprises a new data set all performed in one lab on the same set of inhibitors and enzymes. This includes new assays for approximately 20 additional methyltransferases the results of which have not been reported especially for the older chemical probes. We feel that this new plot is very useful to the community and will save researchers the trouble of looking up and comparing IC₅₀ values and trying to understand if the assay conditions in each paper are comparable to those of another paper.

1d. Figure 3 describes a few new biotinylated derivatives of described tool compounds. These probes are potentially interesting in their own rights, however, for the purpose of this manuscript serve merely as a means to obtain additional selectivity data for probe compounds that were already well characterized with respect to their advertised selectivity from prior publications.

We agree that the derivatized compounds are all interesting in their own right. Some have been previously reported, but 10 here are new. Furthermore, Supplementary **Table 2** indicates where to derivatize those probes for which there is not yet a reported chemical biology reagent. Importantly, while in this manuscript we demonstrate the use of the derivatized probes for chemoproteomics selectivity studies, derivatized probes have a wealth of uses as we point out in our discussion. Moreover, the results of our chemoproteomics studies are not redundant with what is in the literature or Fig. 1C & Supplementary Table 1, and therefore constitute original data and findings. These data provide important cellular selectivity results especially with respect to the ~150 human SAM-dependent PMTs in cells. Furthermore, the chemoproteomics data also demonstrate that the chemical probes and derivatives engage their endogenous protein targets associated interactome in cells, and are therefore, excellent reagents for identifying protein interacting partners/complexes in a cell-type specific manner. Thus, both the derivatives in Supplementary **Table 2** and their use in **Fig. 3** are important components of the Resource.

2. “Figures 4-6 are focused on identifying factors that impact Th1 cell differentiation and they confirm genetic data implicating EZH2/PRC2 in this process. A novel piece of data is provided by the finding that the DOT1L methyltransferase inhibitor promotes Th1 differentiation by increasing the percentage of IFN gamma-positive Th1 cells. While this is an original finding and while DOT1L inhibitor-treated Th1 cells experience hundreds of gene expression changes that distinguish them from control Th1 cells it remains unclear how inhibition of DOT1L affects Th1 biology.”

We agree that the identification of DOT1L as a regulator of Th1 cell differentiation is novel. In this manuscript we demonstrated how the systematic use of the PMT chemical probe collection (Resource) could lead to this novel discovery and its validation. In addition to the screens shown in Figure 4A-D and Figure 6, we confirmed that the increase in the Th1 phenotype upon DOT1L inhibition is coincident with reduction of the H3K79me2 mark (the product of DOT1L; new Fig 4E), and that this mark is dynamically regulated during differentiation of the Th1 lineage. Moreover, the use of the inactive control compound SGC0649 is an important demonstration of the value of using control compounds, and further validates our findings. We feel these results are significant and are a novel application of the probe collection. While the full mechanism underlying the DOT1L-related phenotype remains to be determined, we believe it is beyond the scope of this Resource manuscript.

3. “It also remains unclear whether other lymphocyte subsets are positively or negatively impacted by DOT1L inhibition, an assessment that is necessary to suggest an application as immune-modulating agent for human disease.”

We agree that DOT1L has the potential to be a general regulatory factor in T cells. As described in Figure 6, we demonstrated that inhibition of DOT1L by SGC0946 also enhanced production of IL-17 by Th17 cells, but had no effect on the differentiation of Th2 or iTreg cells. Furthermore, we show the dynamic nature of the H3K79me2 mark during Th1 differentiation and the changes in gene expression associated with DOT1L inhibition in Th1 cells. These are all new and novel results suggesting that DOT1L functions to limit Th1 cell differentiation and maintain lineage integrity.

Thus, overall, DOT1L inhibition appears to be a good candidate as a pro-inflammatory agent, although further studies are needed to fully elucidate this potential. The latter would include the effects of DOT1L inhibition on B cells, macrophages or dendritic cells. We hope such studies will be encouraged by the Resource we provide here. Indeed the Zaph lab is looking into these matters, but such a study is a separate paper in and of itself.

4. “Overall, the manuscript cannot be recommended for publication in Nature Communications since it does not contain sufficient novel data or a coherent piece of data (e.g. around better defining the immune phenotypes upon

DOT1L inhibition) that significantly advances our understanding of how an epigenetic drug could be utilized for immune modulation in human disease.”

We respectfully submit that the main objective of this paper is to provide the detailed description of the PMT chemical probe Resource and demonstrate its use. The detailed elucidation of the mechanism by which DOT1L affects T cell differentiation and how it may contribute to disease modulation is beyond the scope of this Resource paper.

Reviewer #2 (Remarks to the Author):

In "A Chemical Biology Toolbox to Study Protein Methyltransferases and Epigenetic Signaling" the authors compile and validate potency and selectivity information on a well characterized set of methyltransferase probes. They then go on to use this set of 17 probes to rigorously screen for different cellular effects in T cell activation and differentiation. Overall, this paper represents an important resource for the community. In addition, the results with T cells are quite intriguing.

1. *“Comparative western blots (similar to what was done in Figure 4E) for the EED, EZH2/H1, DOT1L and G9a/GLP inhibitor results in the different T cell subsets would be helpful.”*

We agree and have now extended Figure 4E to include the analysis of the specific histone methylation marks (H3K79me2 and H3K27me3) in Th1, Th2, Th17 and iTreg cells in the absence and presence of the chemical probes (and control probes where available). These new results highlight the exquisite specificity of these probes and are discussed in the manuscript.

2. *“As well as the cytokine expression data already presented, it would be helpful to know if any of the T cell subsets from Figure 6 had actual enhanced (or decreased) function due to treatment with different inhibitors.”*

The data presented here represent an example of the usefulness of the chemical probe library. Studies to understand of the precise mechanisms and physiological relevance of each chemical probe in T cell function are beyond the scope of this Resource manuscript.

Reviewer #3 (Remarks to the Author):

The manuscript by Scheer et al. provide comprehensive summary for chemical probes that were developed by the SGC group against multiple of histone methyltransferases. While these chemical probes have been described in previous publications for different biological context, this work is the first comprehensive study for the collection of histone methyltransferase inhibitors on T cell activation and differentiation.

1. *“Experiments in Figure 4 were confusing. The legend said "under Th1 cell condition in the absence (Th0) or presence of indicated probes (1 μM; red) or their controls (where available; black)." Did all probes induce Th0 to Th1 differentiation while positive inhibitors induces IFN-γ? Or all cells used here were actually Th1 cells?”*

We apologise for the confusion. In Figure 4, all data except the Th0 row refers to cells that have been differentiated to Th1 cells after treatment with a probe or control compound. We did not treat Th0 cells with probes. Th0 cells were included as a control to show that untreated Th1 cells produced increased levels of IFN-γ. We have edited the figure legend to clarify this point as follows:

“Fig. 4 Differential Effects of PMT Inhibition on Murine and Human Th1 Cell Differentiation. (A, B) CD4⁺ T cells from the spleen and peripheral lymph nodes of IFN-γ-YFP reporter mice were enriched and polarized

under Th0 (IL-2) or Th1 cell conditions in the absence (Th0) or presence of indicated probes (1 μ M; red) or their controls (where available; black). (A) Flow cytometric analysis of intracellular YFP reporter signal (representing IFN- γ expression) was detected at day 4. (B) Secreted IFN- γ was analyzed by ELISA in the supernatant of the same experiment. Each data point represents one of three biological replicates and the data shown is representative of three independent experiments. (C, D) CD4⁺ T cells from the blood of three healthy human donors were cultured under Th0 or Th1 cell-polarizing conditions in the presence or absence of indicated probes or their controls. (C) Flow cytometric analysis of intracellular IFN- γ was detected at day 4. (D) Secreted IFN- γ was analyzed by ELISA in the supernatant of the same experiment. Each data point represents one of three donors. Dotted lines visualize the mean frequency of IFN- γ -positive Th1 cells in the absence of probes (A, C) or the mean concentration of IFN- γ in the supernatant (B, D). Significant differences are indicated with an asterisk and were calculated using one-way ANOVA (* $p \leq 0.05$, ** $p \leq 0.01$, *** $p \leq 0.001$). (E) Western blot analysis of the effect of indicated inhibitors (red) or control compounds (black) on the trimethylation of H3K27 and dimethylation of H3K79 in CD4⁺ T cells under Th1, Th2, Th17, Treg cell-polarizing conditions. Data shown is representative of 2 independent experiments. See also Supplementary Figs. 2, 3, and 4.”

In addition, we corrected a typographical error in Supplementary Fig. 3A on the x-axis (concentration). Th0 cells were not treated with compound and we corrected the typographical error to reflect this ('1' in the initial submission has been changed to '0').

2. “Expression of signature T cell genes, e.g. T-bet for Th1, GATA3 for Th2 etc. should be provided in addition to cytokine induction.”

This is an interesting and important suggestion. We have now analysed expression of master regulatory factors for Th1 (T-bet), Th2 (GATA3) and Th17 (ROR γ t). Interestingly, we find that expression of lineage-specific transcription factors does not always increase in parallel with enhanced cytokines levels. This suggests that the potential mechanisms of regulation of cytokine expression may be downstream of canonical transcription factor expression. These new data are now presented as new Supplementary Fig. 5.

“Supplementary Fig. 5 Flow cytometric analysis of transcription factor expression in murine Th1 cells activated in the presence of indicated compounds or their corresponding controls. CD4⁺ T cells from the spleen and peripheral lymph nodes of mice were enriched and polarized under Th1, Th2 or Th17 cell-polarising conditions in the presence of indicated probes (1 \$\mu\$ M) or their controls (where available) and analysed for expression of T-bet (Th1), Gata3 (Th2) or ROR \$\gamma\$ t (Th17). Data shown is gated on viable CD4⁺ T cells and is the combined data from 2 independent experiments. Red bars indicate significant downregulation of the transcription factor, black bars indicate significant upregulation. Related to Fig. 6.”

3. *Figure 6 was also confusing. What was the control? What was the reference for fold-change? Was it treatment by negative control compounds? Showing means±S.D might be better way to present the data.*

We apologise for the lack of clarity and have now edited the figure legend. Specifically, the reference for fold-change for each bar is the levels observed with the control compounds (or untreated if no control compound was available).

“Fig. 6. Overview of PMT Inhibition on Murine CD4+ T cell Differentiation. Data is presented as fold-increase (black) or decrease (red) of cytokine production (IFN- γ , IL-13 and IL-17A for Th1, Th2 and Th17 cells, respectively) or FOXP3 expression (Treg cells) over that of the control compound-treated (or untreated) cells. Data shown is from 2 - 3 individual experiments with 3 biological replicates each. See also Supplementary Figs. 5 and 6.”

4. *“The study on T cell differentiation was descriptive. It would be better to include some mechanistic insights on why DOT1L affect Th1 and Th17 specifically while G9a affects mostly Treg etc. Expansion on this point, especially the role of DOT1L in T-cell differentiation, will greatly increase the significance of this study.”*

We agree that the differential effects of PMT inhibition on distinct T cells subsets is a very interesting point. We have now modified the text to include discussion of potential mechanisms that arise from our work, and which could be studied using this Resource in the future.

“One striking finding from our results is that specific probes displayed diverse functional effects on distinct Th cell subsets despite global reductions in their respective specific histone modification in all subsets examined. For example, inhibition of DOT1L led to heightened effector T cell responses with minimal effects on Treg cells, while inhibition of G9a primarily affected Th17 and Treg cells. Although the precise molecular mechanisms are yet to be elucidated, these results suggest that the targeted epigenetic modifiers do not act in isolation but rather collaborate with lineage-specific factors that are expressed in the specific Th cell subsets. As the PMTs do not have sequence-specific DNA-binding domains, Th cell subset-specific accessory factors that target PMTs to the chromatin will be critical modulators of function. In addition, it is possible that non-histone substrates of PMTs are critically important in some Th cell subsets. Thus, this chemical probe library provides tools to examine the fundamental mechanisms associated with epigenetic regulation of Th cells.”

5. *“It would be very useful to include information on whether these probes could be used in vivo.”*

We have now added this information to Supplementary Table 1.

Reviewer #1 (Remarks to the Author):

The revised manuscript by Scheer and colleagues is improved in several aspects. One key issue was the absence of any novel probe compounds in what is advertised as a resource article for the community. The authors now added two new probes (BAY-6035 and SGC3027) to the manuscript which I think is a valuable addition to the manuscript.

The authors also point out that the biotinylated derivatives of already know tool compounds are a valuable contribution given the chemoproteomics data they yield. I tend to agree with this statement, however, I would request that the authors disclose the entirety of the chemoproteomics data in their supplementary material. The current manuscript refers to Figure 3 and Supplementary Table 2 none of which allows one to assess the chemoproteomics data in full.

Regarding the T-cell data the authors have added several data sets that should be useful for the community. While I still believe that the manuscript falls short of capitalizing on the findings of their screen I agree with the argument that the data are interesting to the community as a resource article.

If the authors include the entire chemoproteomics data into the disclosure I now recommend publication.

Reviewer #2 (Remarks to the Author):

The inclusion of western blots in different T cell subsets exposed to inhibitors definitely adds to the paper. I also accept that the intention is for the paper to be a screening resource, so I agree that probing the functionality of the T cells is beyond the scope of the paper. I am satisfied that the authors have answered my concerns.

Reviewer #3 (Remarks to the Author):

The revision by Sheer et al., has addressed some of the previous concerns. However, it seems that there is no significant improvement on the functional and mechanistic characterizations of the

probes related to T cell biology. This work is probably better suited for specialized journals in drug screening.

Reviewer #1 (Remarks to the Author):

1. The revised manuscript by Scheer and colleagues is improved in several aspects. One key issue was the absence of any novel probe compounds in what is advertised as a resource article for the community. The authors now added two new probes (BAY-6035 and SGC3027) to the manuscript which I think is a valuable addition to the manuscript.
2. The authors also point out that the biotinylated derivatives of already know tool compounds are a valuable contribution given the chemoproteomics data they yield. I tend to agree with this statement, however, I would request that the authors disclose the entirety of the chemoproteomics data in their supplementary material. The current manuscript refers to Figure 3 and Supplementary Table 2 none of which allows one to assess the chemoproteomics data in full.
3. Regarding the T-cell data the authors have added several data sets that should be useful for the community. While I still believe that the manuscript falls short of capitalizing on the findings of their screen I agree with the argument that the data are interesting to the community as a resource article.

If the authors include the entire chemoproteomics data into the disclosure I now recommend publication.

“We thank Reviewer 1 for their time to read the original and re-submitted versions of the manuscript, and sharing their knowledge. We agree with Reviewer 1, and apologise for not submitting the entire chemoproteomics data at the original submission. We have added Supplementary Data 1_PRMTi.xlsx, Supplementary Data 2_EED.xlsx, and Supplementary Data 3_DOT1L.xlsx. These contain a prioritized list of the interactors with the protein target.”

Reviewer #2 (Remarks to the Author):

The inclusion of western blots in different T cell subsets exposed to inhibitors definitely adds to the paper. I also accept that the intention is for the paper to be a screening resource, so I agree that probing the functionality of the T cells is beyond the scope of the paper. I am satisfied that the authors have answered my concerns.

“We thank Reviewer 2 for their time to read the original and re-submitted versions of the manuscript, and sharing their knowledge. We are grateful for your acceptance of the re-submitted manuscript.”

Reviewer #3 (Remarks to the Author):

The revision by Sheer et al., has addressed some of the previous concerns. However, it seems that there is no significant improvement on the functional and mechanistic characterizations of the probes related to T cell biology. This work is probably better suited for specialized journals in drug screening.

“We thank Reviewer 3 for their time to read the original and re-submitted versions of the manuscript, and sharing their knowledge. We acknowledge your suggestion to submit the manuscript to a specialized journal in drug screening. However, we feel that more biologists, who are the intended target audience of this Resource on chemical probes, will have the opportunity to read this Resource if published in Nature Communications.”

We hope these final revisions will meet the journal’s requirements for publication and look forward to hearing from you in due course.

Sincerely,

Cheryl Arrowsmith
Structural Genomics Consortium
Professor, Medical Biophysics, University of Toronto
Sr. Scientist, Princess Margaret Cancer Centre
Office: 416 946-0881
Cell: 416 500-5216